

# Impact of improved Sea Surface Temperature representation on the forecast of small Mediterranean catchments hydrological response to heavy precipitation

Alfonso Senatore[1], Luca Furnari[1], Giuseppe Mendicino[1]

1 Department of Environmental and Chemical Engineering, Università della Calabria, P.te P. Bucci 42b, 87036 Rende (CS), Italy

*Correspondence to*: Alfonso Senatore (alfonso.senatore@unical.it)

**Abstract.** Operational meteo-hydrological forecasting chains are affected by many sources of uncertainty. In coastal areas characterized by complex topography, with several medium-to-small size catchments, quantitative precipitation forecast becomes even more challenging due to the interaction of intense air-sea exchanges with coastal orography. For such areas,

quite common in the Mediterranean basin, improved representation of Sea Surface Temperature (SST) space-time patterns can be particularly important. The paper focuses on the relative impact of different accuracy levels of SST representation on regional operational forecasting chains (up to river discharge estimates) over coastal Mediterranean catchments, with respect to other two fundamental options while setting up the system, i.e., the choice of the forcing GCM and the possible use of a three-dimensional variational assimilation (3DVAR) scheme. Two different kinds of severe hydro-meteorological events

affecting the Calabria Region (Southern Italy) on 2015 are analysed using the atmosphere-hydrology modelling system WRF-Hydro in its uncoupled version. Both the events are modelled using the 0.25° resolution Global Forecasting System (GFS) and the ECMWF's 16 km resolution Integrated Forecasting System (IFS) initial and lateral atmospheric boundary conditions. For the IFS-driven forecasts, also the effects of the 3DVAR scheme are analysed. Finally, native initial and lower boundary SST data are replaced with data from the Medspiration Project by IFREMER/CERSAT, having a 24 hour time

resolution and 2.2 km spatial resolution. Precipitation estimates are compared with both ground-based and radar data, as well as discharge estimates with stream gauging stations data. Overall, the experiments highlight that the added value of improved SST representation can be hidden by other more relevant sources of uncertainty, especially the choice of the General Circulation Model providing boundary conditions. Nevertheless, high-resolution SST fields show in most cases a not negligible impact on the simulation of the atmospheric boundary layer processes, modifying flow dynamics and/or the

amount of precipitated water, therefore emphasizing that uncertainty in SST representation should be duly taken into account in coastal areas operational forecasting.

## 1 INTRODUCTION

Operational river flood forecasting is a highly challenging activity for several reasons that go beyond strictly scientific aspects. Hydrometeorological forecasting requires extremely complex systems, where issues like communication of warning,



accessibility of the results and administrative and/or institutional factors can be as important as monitoring and modelling activities (Pagano et al., 2014; Silvestro et al., 2017). Nevertheless, the cornerstone of such systems, and undoubtedly the most demanding part from a scientific point of view, still is the meteorological-hydrological modelling chain, supported by in-situ or remotely sensed measurements.

Increasingly refined modelling chains have been developed in the recent years (e.g., UK Environmental Prediction Research, Canadian Great Lakes, U.S. Navy's Coupled Ocean/Atmosphere Mesoscale Prediction System (COAMPS®)). Despite their complexity, these systems all have to deal with some inherent limitations of the meteorological and hydrological models. The main sources of errors in weather forecast are connected to both inaccuracy in defining the initial state, due to the lack of available measures or observation/assimilation errors, and approximations of the models, whose structure is not capable to
represent properly the phenomena of interest (Allen et al., 2002; Buizza, 2018). These problems are exacerbated by the chaotic nature of the atmosphere. Even though hydrological models are much simpler than meteorological models in their structure (Liu et al., 2012; Pagano et al., 2014), they also have to struggle with different sources of uncertainty that, according to Renard et al. (2010), can be grouped in four categories: 1) input uncertainty; 2) output uncertainty (e.g., runoff estimates are not straightforward); 3) structural model uncertainty and; 4) parametric uncertainty. Furthermore, since very
seldom catchments are perfect natural systems, some effects of human disturbances virtually cannot be modelled.

The main link between atmospheric and hydrological compartments in a forecasting chain is precipitation forecast, which is an output variable for weather models and constitutes the main input for hydrological models. Quantitative Precipitation Forecast (QPF) is a major challenge for operational meteorology, because the reliability of precipitation forecasts crucially affects streamflow forecasts skill (for a review see Cuo et al., 2011; for recent applications, e.g., Davolio et al., 2015; Tao et
al., 2016; Davolio et al., 2017; Li et al., 2017). Among the various strategies adopted for addressing this issue, in the recent years several studies focusing on coastal areas assessed the importance of Sea Surface Temperature (SST) initial and boundary conditions as relevant drivers of QPF, capable to influence, consequently, the streamflow forecast. This impact can be particularly strong in topographically complex coastal areas, characterized by several small catchments, such as in the Mediterranean Basin, for which several research cooperative efforts have been activated (e.g., the MEDiterranean
Experiment, MEDEX, Jansa et al., 2014; the HYdrological cycle in the Mediterranean Experiment, HyMeX, Drobinski et al., 2014).

Several studies focused recently on the effects of sea surface-atmosphere interactions over heavy precipitation at mid-latitudes, particularly in the Mediterranean area (e.g., Manzato et al., 2015; Romaniello et al., 2015; Rainaud et al., 2016). Some of them showed that large variations of the average values of SST boundary conditions significantly affect location
and intensity of high impact events (Lebeaupin et al., 2006; Miglietta et al., 2011; Senatore et al., 2014; Meredith et al., 2015; Pastor et al., 2015; Miglietta et al., 2018; Pytharoulis, 2018). Furthermore, using coupled atmosphere-ocean simulations, Berthou et al. (2014, 2015) highlighted the major effects of long-term SST changes in the representation of Mediterranean intense rain events, even though features at smaller time scales can also contribute significantly. Lebeaupin et al. (2006) found that higher resolution SST fields have poor effects on convection in the case study they analysed (southern



France). Ivatek-Šahdan et al. (2018), examining several events in the Eastern Adriatic, also found that more realistic SST fields did not substantially improve precipitation estimate; furthermore, they showed that the impact of improved SST varied in different cases. Conversely, Katsafados et al. (2011) found noticeable deviations among the forecast skills of simulations with SST boundary conditions at different resolutions in a test-case in the Eastern Mediterranean, while Cassola et al. (2016)

verified in a study in north-western Italy that high resolution SST fields can positively impact QPF in the forecasting range 36-48 h. Finally, Berthou et al. (2016) in southern France and Stocchi and Davolio (2017) in the Adriatic Sea highlighted that SST-atmosphere interactions affect precipitation patterns and intensity mainly through complex (and varying event-by-event) modifications of the stability of the upstream atmospheric boundary layer.

The main objective of this paper is to contribute to the current discussion on the impact of SST representation by extending

the analysis over the whole meteo-hydrological forecasting chain, i.e. going beyond precipitation forecasts and evaluating sensitivity on streamflow forecasts. Furthermore, SST sensitivity is assessed in the context of the overall uncertainty linked to initial and boundary conditions in regional modelling, using different forcing GCMs, with and without data assimilation. To this aim, different accuracy levels of SST representation are used in an operational meteorological-hydrological forecasting chain over a coastal Mediterranean area including, in addition to the native SST fields of the General Circulation

Models (GCMs), also higher resolution fields (namely, the Medspiration level 4 Ultra-High Resolution foundation SST - SSTfnd- from the Medspiration Project by the Centre European Remote Sensing d'Archivage et de Traitement -CERSAT-Institut Français de Recherche pour L'Exploitation de la Mer –IFREMER; ; Merchant et al., 2008; Robinson et al., 2012). Furthermore, two GCM forecasts are used (namely, the Global Forecasting System -GFS- provided by the US National Weather Service -NWS- and the Integrated Forecasting System -IFS- developed at the European Centre for Medium-Range

Weather Forecasts -ECMWF) and a three-dimensional variational assimilation (3DVAR) scheme.

The study area, corresponding to the Calabrian peninsula (southern Italy), due to its particular position in the middle of the Mediterranean Sea and its complex and steep orography experiences quite regularly severe precipitation events and is particularly prone to significant ground effects (Federico et al., 2003a, 2003b; Federico et al., 2008; Chiaravalloti and Gabriele, 2009; Llasat et al., 2013; Gascòn et al., 2016; Avolio and Federico, 2018; up to a very recent flash flood that on

20.08.2018 caused 10 casualties; Avolio et al., 2019). According to Avolio and Federico (2018), severe precipitation events over Calabria can be classified in short-lived events, lasting less than 24 hours, and long-lived events. Following this classification, in this paper two case studies occurred in 2015 are considered, the former characterized by convective, very localized precipitation (August 11-12) and the latter by more persistent and widespread stratiform precipitation (October 30-November 2).

The meteorological-hydrological forecasting chain is based on the WRF-Hydro modelling system (Gochis et al., 2015). This open-source community model, originally developed as the hydrological extension of the Weather Research and Forecast (WRF) model, provides a coupling architecture allowing to connect vertical water fluxes between the earth surface and the atmosphere, simulated at coarse resolution by the atmospheric model, to lateral surface and sub-surface fluxes simulated at high resolution by the hydrological model, both in one-way (i.e., with no feedback from the routing models to the





atmosphere) and two-way (with feedback) manner. WRF-Hydro system dramatically evolved in last years (Salas et al., 2018; Lin et al., 2018: Lahmers et al., 2019), being operationally adopted into the NOAA National Water Model (NWM) across the continental U.S, besides being used for research applications (e.g., Yucel et al., 2015; Senatore et al., 2015; Arnault et al., 2016; Verri et al., 2017).

5  The paper is organized as follows. Section 2 describes the study area, the two events analysed, the numerical model and its setup with details on space and time resolutions of the boundary conditions. In Section 3 the results of the meteorological and hydrological outputs are analysed separately for the two events. Finally, Section 4 discusses and summarizes the main findings and outlines future research lines.

## 2. MATERIALS AND METHODS

### 2.1 Study area and events description

Location of Calabrian peninsula in the centre of the Mediterranean as well as its complex orography entails a very irregular precipitation distribution (average annual precipitation varies between 600 and 1500 mm; Federico et al., 2010) and amplifies the occurrence of extreme weather events, which have often caused deaths (about 200 in the period between 1980

and 2016; Petrucci et al., 2018). Among the relatively numerous recent events, this study focuses on two case studies occurred in 2015 and characterized by distinctive different features.

The first high impact event (case study 1) was very localized in space and time and hit the north-eastern part of the region on the morning of 12 August 2015. The analysis at the synoptic scale (Figs. 1a-f) shows that in the early hours of 12 August 2015 a main low pressure system coming from the Atlantic moved over the French and Spanish coasts, while over the

central Mediterranean a cut-off low occurred, giving rise to a new low pressure vortex with reduced dimensions that caused intense local rainfall. The observed precipitation patterns (Fig. 1g) involved only small areas in the mainland, specifically the territory of the Corigliano and Rossano municipalities. The data provided by the Italian National Radar Network (integrated in the same map of Fig. 1g), though underestimating ground observations, show that most of the precipitation occurred over the Ionian Sea. The Corigliano rain gauge measured high rainfall values (Fig. 1h). During the 48 hours from 00:00 UTC 11

August 2015 up to 00:00 UTC 13 August 2015, 255.2 mm of rain were recorded, with a maximum of 246.4 mm in 24 hours (from 6:00 pm 11 August to 6:00 pm 12 August), 223.2 mm in 12 hours (from 01:45 am 12 August to 01:45 pm 12 August), 167.4 mm in 6 hours, 107.2 mm in 3 hours and 51.4 mm in 1 hour. The hydrological impact concerned some small/very small coastal catchments, the most important of which is the Citrea Creek (11.4 $km^2$, catchment boundaries highlighted in Fig. 1i), which overflowed causing several tens of millions of euros of damage.

The second event (case study 2) involved a much larger area and developed over 4 days, from 30 October to 2 November 2015. The synoptic analysis (Figs. 2a-c) shows another cut-off low, remaining stationary over Sicily for much of the period and attracting humid and warm air from the Ionian Sea to the southeast (a detailed synoptic description of the event is





provided by Avolio et al., 2018). The orographic effect in this event turned out to be decisive, the Calabrian mountain ranges acting as a real barrier, therefore large part of the rainfall occurred in the Ionian (eastern) side of the region. While on 30 October 2015 only the northern part of the region was affected (Fig. 2d; about 200 mm in 24 hours in the Oriolo station), the highest precipitation during the entire event was recorded in the southern coast (Figs. 2e-g), with a maximum of about 740

5 mm (Chiaravalle Centrale station) and a daily maximum of about 370 mm (Sant'Agata del Bianco). In Figs. 2d-g, rain gauges observations overlap the precipitation fields detected by the weather radars, extending also over the sea. The hydrological impact of the event concerned the whole eastern side of the region. Two catchments are selected for this study, namely the Ancinale River closed at the Razzona gauging station (116 km$^2$, Fig. 2h) and the Bonamico Creek closed at the Casignana gauging station (138 km$^2$, Fig. 2i). Such catchments are chosen because they are two of the biggest with available

observations of water levels (unfortunately no discharge data are available) and are located in the north and the south, respectively, of the rainiest area. Specifically, Chiaravalle Centrale station is located at the Ancinale River outlet.

## 2.2 Numerical model description and setup

### 2.2.1 WRF

The Advanced Research WRF (ARW) Model, version 3.7.1, is used in two one-way nested domains (Fig. 3). The external

domain D01 covers a large area of the Mediterranean (33.04°-49.85°N, 3.59°-28.59°E) with a 10 km (187 × 205 grid points) horizontal resolution, while the innermost domain D02 is centred over the Calabrian peninsula (37.10°-40.87°N, 13.88°-18.71°E), with a 2 km (200 × 200 grid points) horizontal resolution. The model runs on 44 vertical atmospheric layers, up to a 50-hPa pressure top (about 20000 m), and on 4 soil layers, down to 2 m below the surface. The time step of the model simulation is 60 s in D01 and 12 s in D02.

Physical parametrization of the model is the same used by Senatore et al. (2014) and is reported in Table 1. Boundary and initial conditions are provided by two operational forecast GCMs, namely the GFS in forecast mode with a spatial resolution of 0.25° and the IFS-ECMWF High RESolution (HRES) in forecast mode with a spatial resolution of about 16 km. In both cases boundary conditions are provided each 6 hours. As a further step, both initial and lower boundary SST data are replaced with the Medspiration L4 Ultra-High Resolution SSTfnd (obtained as daily mean with a resolution of 0.022°). The

high resolution Medspiration SST fields are ingested through GIS-based techniques into the WRF initial and lower boundary conditions files of both domains following Senatore et al. (2014). Furthermore, two relevant options allowed by the WRF modelling system are always activated for all SST boundary conditions: the *sst_update* option, allowing dynamical lower boundary (i.e., SST) conditions, and the *sst_skin* option, based on Zeng and Beljaars (2005), which permits the simulation of SST dynamics.

Initially, also the European Ocean-Sea Surface Temperature MultiSensor L4 Three-Hourly Observations provided by the Copernicus Marine Environment Monitoring Service (CMEMS), offering higher time and space (0.02°) resolution, were considered, nevertheless such products display occasional but serious quality degradation (personal communication from





CMEMS Service Desk) that in both our case studies revealed a checkerboard effect when looking at the SST values near the Italian coasts of the Adriatic Sea.

Finally, both as an additional comparison and with the aim of highlighting its relative impact with respect to the effects of different boundary conditions provided by different GCMs and/or improved SST fields, also a data assimilation technique is

used for both the test cases. Specifically, a 3DVAR assimilation scheme (Barker et al., 2004; Huang et al., 2009; Barker et al., 2012) is adopted, introducing conventional meteorological observations into the initial conditions and adjusting boundary conditions to improve simulations performance.

A summary of all simulations carried out is reported in Table 2.

**2.2.2 WRF-Hydro**

In this work, WRF-Hydro version 3.0 is used in one-way mode. Therefore, the atmospheric model outputs are used as input of the hydrological model using an hourly time step. According to the WRF parameterization, the Land Surface Model (LSM) is Unified Noah and is used at the same resolution of the domain D02, while for the lateral routing of surface and subsurface water an increased horizontal resolution of 200 m is used (2000 × 2000 grid points), thus having an aggregation

factor of 1 / 10 from the atmospheric to the hydrological model.

No observed discharge or flow depth data is available for case study 1, hence model calibration is not performed. In case study 2, model calibration is performed manually with respect to the available water level data for the two selected catchments (Ancinale and Bonamico), with the aim of reproducing the timing of the hydrological responses to heavy precipitation and, mainly, to correctly simulate the peak flow time, which is a paramount variable for civil protection

activities.

The humidity and temperature conditions in the 4 soil layers at the beginning of the analysed event (30 October 2015 00:00 UTC) are achieved through off-line simulations with a spin-up time of one month. The meteorological forcing for this period is basically given by the spatial interpolation of ground-based observations (provided by the monitoring network managed by the Centro Funzionale Multirischi-ARPACAL, Calabria Region). The interpolation techniques adopted are the same

described in Senatore et al. (2015) except that precipitation fields are interpolated through Inverse Distance Weighting (IDW) instead of exponential kriging. Furthermore, only during the event (i.e., from 30 October to 2 November 2015) precipitation fields are achieved merging hourly ground-based rainfall observations to hourly radar data estimates provided by the Italian weather radar network managed by the National Department of Civil Protection. The merging procedure follows Sinclair and Pergram (2005) with the difference that, instead of a double kriging interpolation, a simpler double IDW

interpolation method is used. The merging technique guarantees an increase of the total "observed" rainfall volume, with respect to a simple IDW interpolation, of +4.6% over the Ancinale River and +10.6% over the Bonamico Creek.

The parameters involved in the calibration procedure are broadly the same used in previous studies with WRF-Hydro (e.g., Yucel et al., 2015; Senatore et al., 2015). Specifically, the LSM parameters calibrated are the infiltration factor (REFKDT),



the coefficient governing deep drainage (SLOPE) and the thicknesses of the 4 soil layers. In addition, two spatially distributed parameters of the hydrological model, namely the overland flow roughness scaling factor (OVROUGHRTFAC) and the initial retention height scaling factor (RETDEPRTFAC), are calibrated together with the Manning roughness coefficients (one value for each stream order).

The calibrated parameters are shown in Table 3, while resulting hydrographs are shown in Fig. 4. The more impulsive behaviour of the Bonamico Creek, typical of Calabrian "fiumare", is simulated through lower values of the infiltration factor and lower soil layers thickness. Nevertheless, in order to allow timely peak flows simulation, a small delay of the initial response is necessary through an increase in the RETDEPRTFAC value, which is compatible with noteworthy initial ponding in the wide alluvial bed and infiltration in the gravelly soil. On the other hand, abundance of organic matter in the
soils of the dense forests within the Ancinale River catchment, which especially in autumn can store considerable quantities of water, most probably contributes substantially to the smoother response of the Ancinale River.

As for the hydrographs (Fig. 4), adopting typical stage-discharge power relationships (i.e., $q = a \cdot h^b$, where q is the discharge, h the water level and a and b two calibration coefficients) the coefficients of determination ($R^2$) between simulated discharge values and observed water levels are equal to 0.942 and 0.831, respectively for the Ancinale River and
the Bonamico Creek. Concerning the reliability of the simulated discharge amount, since reference observations are missing, an indirect validation of the peak flows achieved is performed using the Hydrologic Engineering Center's (CEIWR-HEC) River Analysis System (HEC-RAS) (Hydrologic Engineering Center, 2016). Cross-sections, for both the outlets of the catchments and for 4 upstream and downstream points approximately spaced 50 m, are determined merging data from a ultra-high resolution (5 m) Digital Terrain Model provided by the Calabria Region Cartographic Centre with the heights
given in very recent official maps (Technical Cartography of Calabria Region) at a scale of 1:5000. Such cross-sections are further validated by on-field sample measurements. One-dimensional steady flow simulations reaching observed peak heights provide peak discharges broadly comparable to the results achieved with the model.

For the sake of brevity, hereafter the WRF-Hydro hydrographs calibrated using observed precipitation fields shown in Fig. 4 will be referred to as 'observed hydrographs' or simply 'observations'.

**3. RESULTS AND DISCUSSION**

**3.1 Case Study 1**

Figure 5 shows the skin SST evolution for two specific points 1 and 2 in the Ionian Sea (whose exact location is given in Fig. 3b) for the whole 48-hour simulation period (from 11 August 00:00 UTC to 13 August 00:00 UTC). Furthermore, panels in Figure S1 show, for all simulations carried out in this case study, the skin SST fields in the Domain D02 from 11 August
18:00 UTC to 12 August 18:00 UTC with a time step of six hours.

The main features highlighted by the skin SST maps are the strong underestimation of native IFS fields close to the coastline (this is due to a known interpolation problem along coastlines that lowers temperatures to unrealistic values; L. Magnusson,





personal communication) and the overestimation, especially off the Tyrrhenian Sea (up to more than 2 K), of the native GFS fields. The other skin SST fields mostly differ each other less than ±0.5 K. It is noteworthy that skin SST fields in the simulations using Medspiration product are not identical due to the fact that, with the method of Zeng and Beljaars (2005), skin SST values are influenced by the surface winds and net radiation fluxes modelled by the different simulations.

Because of the IFS underestimation near the coastline, a comparison of the skin SST average hourly evolutions among simulations in the whole domain D02 would be not completely explicative. For this reason, focusing on the Ionian Sea, in Fig. 5 the points 1, closer to Corigliano-Rossano, and 2, off the Calabrian southern coast, are examined. Concerning daily values, a clear but slight (<0.5 K) overestimation of GFS-O is shown for both days in point 1 and on the second day in point 2. Some hourly differences are more evident: e.g., in point 1 GFS-O values are up to 1.5 K higher than other models on 12

August at around 12:00, while in point 2 peak values of IFS-O and IFS-M on 11 August at around 12:00 are about 1 K higher than other models. Nevertheless, the differences among models during the night between 11 and 12 August (i.e., right before and during the rain event) are generally low. The only noteworthy difference is given, in point 2, by the small underestimation of Medspiration simulations (i.e., GFS-M, IFS-M and IFS-DA-M) of about 0.3-0.4 K, given by a sudden reduction of their skin SST values, most probably due to the change of the Medspiration SST field (from 11 August to 12

August). This behaviour, clearly not realistic, highlights a weakness occurring while ingesting directly such external data in the WRF simulation.

The accumulated precipitation modelled by all simulations for the 24-hour period from 11 August 18:00 UTC to 12 August 18:00 UTC is shown in Figure 6. Overall, all models miss the location of the event, moving it further south, off the Ionian coasts. GFS-based simulations forecast more rainfall than IFS-based (average values in the domain of 10.1 mm and 8.9 mm

with the native SST fields, 10.4 mm and 9.5 mm with the Medspiration SST fields, respectively for GFS and IFS), but more centred to the south. IFS-based simulations forecast rainfall clusters with more elongated shapes in the south-north direction, allowing more precipitation to reach the central-northern Ionian coasts (namely, the Corigliano-Rossano area). Though simulations based on the 3DVAR scheme still miss the correct location of the event, they both provide more rainfall in the domain (average values of 11.2 mm with the native SST fields and 11.0 mm with the Medspiration SST fields) and show

also a well-defined rainfall cluster close to the central Ionian coast. Both IFS and 3DVAR simulations overestimate land precipitation in that area.

According to the generally small differences identified in the SST fields, Figure 6 clearly shows that ingesting high resolution SST information provides, in terms of spatial distribution of accumulated precipitation, much less relevant (and partially chaotic) effects than changing initial and boundary conditions or using data assimilation schemes, and a minor or

possibly opposite impact on the accuracy of the simulations. Given the peculiar features of the analysed event, it makes sense to focus on the area surrounding the Corigliano gauge station. For each simulation, the graph in Figure 7a merges intensity, location and time correlation information of the closest rainfall peaks (with a threshold of at least 40 mm) to that station, while Figure 7b explicitly shows the time evolution of accumulated rainfall for each of the locations identified (the points are highlighted with small stars in the panels of Figure 6). Given that all simulations strongly underestimate the





observed rainfall value of 246.4 mm (the highest simulated value of about 100 mm is given by IFS-DA-M), there is no configuration clearly over-performing the others. Both GFS peaks are located to the south (about 20 km) and delay the rain event from 8 (GFS-M) to 11 (GFS-O) hours. IFS-O and IFS-DA-O peaks are lower than IFS-M and IFS-DA-M, but they are generally closer to the Corigliano station (about 13 km and 22 km, respectively). Furthermore, Figure 7b shows that

ingesting Medspiration fields moves up the rainfall events for both IFS-O and IFS-DA-O. This suggests that removing the unrealistic low SST values along the coastline near the Corigliano station (i.e., considering IFS-M and IFS-DA-M in place of IFS-O and IFS-DA-O, respectively) produces the double effect of increasing rainfall amounts and accelerating flow dynamics. Such effect is more easily recognizable looking at the 3DVAR simulations, which provide more water vapour and precipitation. Figure 8 highlights that, moving from IFS-O to IFS-DA-O to IFS-DA-M, 850 hPa wind speed on 12 August at

00:00 UTC generally increases in domain D02 and specifically off the northern Ionian coasts of Calabria. As a result, Figure 9 shows that, moving from IFS-O to IFS-DA-O to IFS-DA-M, the integrated water vapour (IWV) cluster off the Ionian Sea simulated three hours later (03:00 UTC) is both bigger and closer to the coast.

Differences between IFS-O and IFS-DA-O are due to the assimilation in the domain D01 of 14 vertical profiles of pressure, wind speed and direction, absolute and dew point temperature and relative humidity, together with 14 point measurements

provided by aircrafts at a fixed pressure level (corresponding to about 12 km). Instead, differences between IFS-DA-O and IFS-DA-M are mainly due to different skin SST values. Specifically, higher SST values given by ingesting Medspiration fields on the one hand enhance water vapour concentration in the atmosphere (the average upward moisture flux from sea surface in domain D01 increases of about 5%), on the other hand affect the stability of the atmospheric boundary layer, providing more energy to the system and accelerating the flow dynamics (such as found, e.g., by Stocchi and Davolio, 2017).

The early arrival of the moist air mass towards the area of Corigliano-Rossano using Medspiration SST fields is highlighted by the time series of hourly averaged water vapour flux through section CC' (Fig. 10, with the Section shown in Fig. 3b). Local flow peaks are moved up from 2 to 4 hours in advance with IFS-DA-M compared to IFS-DA-O, and a similar behaviour, even though less evident, is observed with IFS-M compared to IFS-O.

Concerning the assessment of the hydrological impact of the forecasted event, notwithstanding the detailed analysis

performed, case study 1 does not provide relevant results. The centre of the Citrea catchment is located 8 km approximately south-east of the Corigliano gauge station, has a maximum length of about 7 km in the south-north direction and a maximum width of only 2.5 km. The level of accuracy achieved by all simulations performed is not yet enough to correctly forecast the hydrological impact for such small catchments in areas with very complex topography like that under study. The maximum rainfall accumulated value over the catchment is forecasted by IFS-O, with 16 mm in 3 hours. However, the accuracy of the

models is already high enough to make them very useful (it is worthwhile to recall that the starting time of the simulation is more than 24 hours before the event). In fact, if model forecasts are used for inferring information about wider 'warning areas' than single small catchments (such as the Italian Civil Protection system actually does), they provide essential inputs for civil protection activities.



### 3.2 Case Study 2

Case study 2 embraces a longer period than case study 1. In this Section, forecasting skills are first assessed considering the whole 4-days length of the event. Then, in order to reduce the uncertainties due to the longer lead time forecast, we focus on a 3-days forecast, starting on 31 October 2015.

### 3.2.1 4-day forecast (30 October – 2 November 2015)

Such as in the previous case study, the first analysis is devoted to skin SST fields. Fig. S2 highlights (besides the already mentioned IFS-related problem along coastlines) that in this case Medspiration fields for the whole period overestimate both GFS and IFS native SST fields. Specifically, average differences with respect to GFS SST vary from about 0.6 to 0.8 K, while differences with respect to IFS SST fields are higher than 0.8 K (the average difference increases to about 1.5 K, if

also the values along coastlines are considered). It is noteworthy that also GFS underestimates skin SST particularly near coastlines, while, such as in the previous test case, there is an overestimation off the Tyrrhenian Sea. Focusing on points 1 and 2 (Fig. 11) it is shown that: 1) both points replicate the general behaviour, with Medspiration fields values higher than GFS, in their turn higher than IFS; 2) differences are more marked in point 1 (average values of +1.0 and +0.6 K, respectively for IFS and GFS) than in point 2 (+0.9 and +0.3 K); 3) such as in the case study 1, also here, in the graph related

to point 1, a sudden reduction of about 0.5 K can be observed for Medspiration, moving from 1 November to 2 November. Nevertheless, a similar abrupt change, even though less marked (about 0.2 K), is observed also for GFS on 31 October 06:00 UTC. Summarizing, this case study shows an evident skin SST increase from IFS to GFS to Medspiration.

Figure 12 shows the accumulated rainfall fields in the 4-day simulation period by the 6 WRF configurations compared with a rainfall map of Calabria achieved merging ground measurements to radar observations (the merging procedure followed

Sinclair and Pergram, 2005; distinct rain gauge and radar data are available in Figs. 2d to 2g). It clearly highlights, in agreement with the previous case study, that the main impact on rainfall output is given by the choice of the GCM providing boundary conditions. Average accumulated precipitation in domain D02 is equal to 80 mm with GFS-O, 71 mm with IFS-O and 68 mm with IFS-DA-O. Interestingly, the introduction of the 3DVAR scheme this time leads to reduced precipitation (in the domain D01 22 vertical profiles and 16 points measurements are assimilated). Higher skin SST values with Medspiration

result in increased average precipitation in the domain D02 for all three cases, from +8% (IFS-DA) to +11% (GFS). Concerning the precipitation patterns, for the aims of this study it is interesting to focus on the biggest cluster in the south-east corner of the domain (i.e., the direction from which the humid air mass comes). Moving from GFS to IFS to IFS-DA, quite independently from SST fields change, a shift of this cluster can be observed from north-east to south-west.

With the aim of objectively assessing the performance of each WRF configuration, a detailed analysis using categorical

scores is carried out considering ground based observations in the Civil Protection warning areas more affected by the event (grey areas in Figure 13a). Specifically, 30, 19 and 22 rain gauges are considered, respectively for zones Cala4, Cala7 and





Cala8. Among the numerous scores available in literature (for a review see, e.g., Wilks, 2006), for each zone Figure 13 shows results concerning the Frequency Bias Index (FBI):

$$FBI = \frac{hits + false\ alarms}{hits + misses} \tag{1}$$

and the Equitable Threat Score (ETS):

$$ETS = \frac{hits - hits_r}{hits + misses + false\ alarms + hits_r} \tag{2}$$

where

$$hits_r = \frac{(hits + misses)(hits + fase\ alarms)}{hits + misses + false\ alarms + correct\ negatives} \tag{3}$$

In the previous equations, the terms *hits*, *misses*, *false alarms* and *correct negatives* refer to a typical $2 \times 2$ contingency table. The FBI indicates if the forecast system has a tendency to underestimate (FBI<1) or overestimate (FBI>1) events frequency,

while ETS measures the fraction of the correctly predicted events, adjusted for hits associated with random forecasts, and ranges from -1/3 to 1 (perfect score). Both scores are used for consecutive 6-hour time intervals for the period 31 October -2 November (which is the actual rainy period for the analysed warning areas), using precipitation thresholds with a step of 0.2 mm from 0.2 to 1 mm, a step of 1 mm up to 10 mm, a step of 2 mm up to 20 mm and a step of 5 mm up to 50 mm.

ETS graphs show the generally better performance of IFS-DA-M, especially for higher thresholds. Other models have

15 discording levels of accuracy: e.g., IFS-DA is the best in Cala4, but the worst in Cala7. Nevertheless, ingesting high resolution SST generally provides better scores in all cases. Complementary information provided by FBI highlights significant underforecast of GFS-based simulations in both Cala4 and Cala8 and overforecast in Cala7. Other simulations behave better, but also FBI points out that the 3DVAR scheme alone does not necessarily improve IFS-based forecasts (e.g., in Cala7 IFS-O is more accurate than IFS-DA-O), unless also an improved SST representation is considered (IFS-DA-M

always shows FBI values around 1). Results achieved with ETS and FBI are generally confirmed also by other scores not shown, such as the Probability of Detection (POD) score or the False Alarm Rate (FAR).

As stated previously, higher skin SST Medspiration values affect precipitation magnitude. This outcome agrees with the average increase of upward moisture flux from the sea surface in domain D01 (+8% with GFS, +13% with IFS and IFS-DA). Vice versa (and contrary to what was found in the previous case study) the simulations do not show relevant differences in

the timing of the event. If the accumulated values of average precipitation in each of the warning areas are considered, all simulations are very highly correlated (≥0.98, graph not shown) with observations. Fig. 14, showing the time series of hourly averaged water vapour flux through section DD', highlights that there are no relevant either forward or backward time deviations between the simulations with original skin SST fields and the corresponding simulations with Medspiration fields. The main effect observed in Fig. 14 is the lower flux of the GFS-based simulations, because the main flow of soil moisture is

shifted towards north-east with respect to the section DD' (in agreement with the precipitation maps in Fig. 12). The average flux increase with IFS-M and IFS-DA-M is of about 3-4%, vs. IFS-O and IFS-DA-O, respectively. Fig. 15, showing a snapshot of the IWV distribution in D01 during the event (31 October at 21:00 UTC), confirms the similar timing of the simulations. Moving from IFS-O to IFS-DA-O to IFS-DA-M, the size of the cluster of humid air south of Calabria increases,


but its position is substantially the same. Fig. 16 provides additional information about 850 hPa wind fields in D02 at the same time of Fig. 15. The area in the south-east corner of the maps with strong south-easterly winds increases its width moving from IFS-O to IFS-DA-M, but the wind fields spatial patterns are rather similar (with even weaker winds off the Ionian Coast of Central Calabria with IFS-DA-M).

All simulations performed for this case study show that the greater energy supplied to the system by the higher skin SST Medspiration fields affects lower layers flow dynamics allowing more transport, but not accelerating it. This behaviour can be attributed to the long-lasting characteristics of the event that, developing at a wider scale than case study 1 and providing humid air continuously, smooths potential differences in terms of timing.

Assessing the hydrological impact in the two selected catchments is more interesting in this case study, because all
simulations forecast heavy rain over the catchment areas of the Ancinale River and Bonamico Creek, yet it is still challenging, because reliable hydrological forecasts require accurate QPFs at the catchment scale. A QPF performance analysis is carried out for the catchment areas, considering the average values of the interpolated precipitation fields. The simulated average precipitation over the Ancinale River catchment is strongly overestimated by all the IFS-based simulations (from +53% to +72%), while GFS-based simulations provide much more reasonable biases (+12% and -1%,
respectively for GFS-O and GFS-M). Concerning the Bonamico Creek catchment, IFS-based biases are smaller in general (from -13% to +22%, but +54% with IFS-DA-M), while GFS-based simulations strongly underforecast (-56% and -43%, with GFS-O and GFS-M respectively). Taylor diagrams in Figs. 17a and 17c, based on the comparison of the average rainfall time series, generally confirm the results of the bias analysis, with better performances of the GFS-based simulations in the Ancinale River and of the IFS simulations in the Bonamico Creek.

QPF analysis only partially reflects the main outcomes of the hydrological simulations. According to precipitation overforecast of the IFS-based simulations, both Figs. 17b and 17d show a very relevant overestimation of all IFS-based hydrographs (except IFS-M in the Bonamico Creek). Nevertheless, in the case of the Ancinale River, IFS-O, IFS-M and IFS-DA-M hydrographs are reasonably correlated with observations (correlation coefficient $r$ equal to 0.62, 0.77 and 0.70, respectively). Furthermore, simulated peak flow times of IFS-O and IFS-M hydrographs are very close to the observed
occurring on 1 November 12 UTC (1 hour before and 4 hours after, respectively). The hydrographs resulting from GFS simulations are closer to observations in terms of volumes, nevertheless peak times are significantly anticipated or delayed. Concerning the Bonamico Creek, IFS-based hydrographs are not well correlated and forecast peak flows more than 12 hours in advance with respect to observations, while GFS-based hydrographs substantially underestimate.

The analyses performed show the great uncertainty of QPF at the catchment scale, due to many sources of errors and
uncertainties that can be amplified by the 4-day forecast window. In order to attempt to reduce the sources of uncertainty and highlight the possible emergence of positive effects due to the more detailed representation of the SST, a further analysis is carried out with a forecast window reduced to 3 days, from 31 October to 2 November.





### 3.2.2 3-day forecast (31 October – 2 November 2015)

The main change produced by the 3-day forecast with respect to the 4-day forecast is the higher correspondence of the GFS-based simulations to the IFS-based. Specifically, Fig. 18 highlights that the GFS-based rainfall footprints located in the south-east of the domain D02 meet the Calabrian Ionian coast more southern with respect to the 4-day simulation (Fig. 12), in agreement with the IFS-based simulations. Overall, the simulated rainfall fields are rather similar to each other and seem to reproduce reasonably well the observations in the southern part of the region (i.e., the area most affected by the event), while overforecast is noted in the central zone, such as it was found also in the 4-day simulations. 3DVAR forecasts starting on 31 October assimilate 15 vertical profiles and 12 point measurements. Though the IFS-based simulations forecast higher rainfall peaks off the southern Ionian coast (up to 1000 mm), the average accumulated precipitation in D02 is almost identical for all simulations (51 mm with GFS-O, 52 mm with IFS-O and 53 mm with IFS-DA-O). Precipitation increase caused by the higher skin SST Medspiration fields varies from +9% (IFS-DA) to +12% (GFS), in agreement with the upward moisture flux increase in D01 (+7% with GFS, +8% with IFS and IFS-DA).

Performance evaluation with categorical scores against ground based observations is repeated also for the 3-day simulations with the same warning areas Cala4, Cala7 and Cala8 (Fig. 19). Despite the improvements with respect to the 4-day simulations, the ETS values of the GFS-based simulations are generally lower and introducing the Medspiration fields does not help to improve their performance. On the other hand, a more detailed SST resolution increases the ETS values of the IFS-based simulations in zones Cala4 and Cala8 (but not in zone Cala7). Concerning bias, FBI graphs show substantial underforecasts in zones Cala4 and Cala8 and overforecast in zone Cala7. However, in this case the GFS-based simulations provide better results, especially in zone Cala7 and for high thresholds.

Focusing on the QPF assessment at the extent of the study catchments, the bias analysis does not show decisive improvements with respect to the 4-day forecasts. In the Ancinale River catchment, IFS-based simulations overforecasts are reduced to about +40% (except an increase to +80% with IFS-O), GFS-M still provides a nearly unbiased estimate (-3%) but GFS-O overforecast worsens to +44%. In the Bonamico Creek catchment, 3-day bias is reduced for IFS-DA forecasts and slightly increased for IFS forecasts, with values however around -20% in all cases. GFS-based forecasts are much improved (+10% and -16%, for GFS-O and GFS-M respectively). Taylor diagrams (Figs. 20a and 20c) largely confirm the outcomes achieved with the 4-day analysis, i.e. better performance of one of the two GFS-based forecasts (GFS-M) in the Ancinale River catchment and of the IFS-based forecasts in the Bonamico Creek catchment.

The hydrological simulations over the Ancinale River (Fig. 20b) are affected by precipitation overforecasts for what regards discharge values. Furthermore, all simulations forecast the peak flows in advance with respect to observations. Nevertheless, IFS-M, IFS-DA-O and IFS-DA-M show $r$ values around 0.6. In particular, IFS-DA-O (highest $r$, equal to 0.65) forecasts the peak flow only 4 hours in advance. Despite the good performance with precipitation forecasts, GFS-M hydrograph is not well correlated with observations and simulates the observed peak flow about 9 hours in advance. With the Bonamico Creek IFS-DA-O results are even better (Fig. 20d). The simulated peak flow, according to precipitation forecasts, underestimates





the observed of about 20%, but the correlation between simulated and observed hydrographs is high (0.89) and the observed peak flow time (1 November 16 UTC) is delayed by only 2 hours. Generally, all the IFS-based simulations are well correlated (*r* values always higher than 0.6) even though peak flow time is always delayed (up to 12 hours). GFS-based simulations are poorly correlated and show significant overestimation and early forecast of the peak flow.

**4. DISCUSSIONS AND CONCLUSIONS**

The results achieved in this study provide not univocal indications and need to be carefully analysed. Table 4 aims at supporting the discussion summarizing the main outcomes concerning: 1) representation of the skin SST fields; 2) accumulated precipitation values in the internal domain and the related spatial distribution; 3) time distribution of precipitation and; 4) hydrological impact (hydrograph shape, total discharge, peak flow times), depending on: 1) GCM

choice for determining the boundary conditions; 2) use of the 3DVAR scheme; 3) use of the high-resolution Medspiration fields.

The most evident outcome across the case studies, yet far from surprising, is that the choice of the GCM providing boundary conditions is, comparatively, the most relevant factor affecting the simulations. Specifically, for the case studies analysed, GFS-based simulations are generally less performing than IFS-based (this difference is emphasized if the forecast time

window is increased, such as case study 2 demonstrates). Of course, it is not a generalizable result, given the few number of events involved and the lack of further analyses (e.g., evaluation of different parameterizations). For example, for case study 2, through detailed sensitivity tests Avolio et al. (2018) found that simulations forced by GFS have better performance than those forced by ECMWF. Nevertheless, for the purpose of this study it is shown that the different features differentiating the two GCMs (among that, the spatial resolution, which is improved with IFS) can considerably affect precipitation fields

calculated through dynamical downscaling, comparatively more than using three dimensional variational data assimilation methods or imposing specific (high-resolution) skin SST boundary conditions.

The use of the 3DVAR scheme in this study has to be considered mainly as a strategy for improving initial conditions. Several studies adopted data assimilation approaches for achieving improvements for shorter forecast periods than 48 to 96 hours used in this study (e.g., Sun et al., 2106; Gustafsson et al., 2018; Thiruvengadam et al., 2019), unless specific strategies

were used (e.g., with cycling 3DVAR runs; Liu et al., 2018). Here, we focus on 2- to 4-days periods (depending on the case studies) for the sake of simplicity and clearness, testing the capability of different model configurations to reproduce the overall development of the hydrometeorological events, from their beginning to their end, checking also their usefulness in providing proper warning lead time. Therefore, testing extensively the 3DVAR scheme in order to get the highest benefit goes beyond the aims of this study.

Even though used with the underlined limitations, 3DVAR simulations provide some worthy outcomes. Concerning case study 1, applying the 3DVAR scheme with IFS boundary conditions results in a substantial increase of the average rainfall in the innermost domain (up to 25%), but this change does not provide clear advances in forecasting skills. That is consistent



with previous studies, demonstrating that the effects of data assimilation do not lead to an effective improvement in the case of highly convective events (Liu et al., 2013). In case study 2 it is noteworthy that IFS-O is capable to provide in some warning areas, especially in the 4-day forecasts, better ETS values than IFS-DA-O, meaning that other sources of uncertainty than initial conditions can strongly affect forecasting skills. Among those uncertainties, representation of SST conditions can

be important, given that, in general, IFS-DA-M (i.e., the simulation including both data assimilation and improved SST representation) provides better performances.

Unlike the 3DVAR scheme, the effects of improved SST representation on forecasts are emphasized to the maximum in this study, given that observed rather than forecasted SST fields are replaced as lower boundary conditions in the simulations, thereby providing a kind of "upper limit" to the effects provided by well forecasted SST fields. The foundation SST fields

used (defined as the temperature of the water column free of diurnal temperature variability, Donlon et al., 2007) are produced by the Medspiration project once every 24 hours, but the diurnal cycles are ensured by the *sst_skin* option. They especially improve the SST fields provided by IFS boundary conditions that, even allowing better forecasts than GFS, show very evident problems along the coastlines. The forecast periods analysed in this study allow to largely overcome the problem highlighted by Cassola et al. (2016), who found that for forecasting ranges shorter than 36-48 hours the forced

ingestion of high-resolution SST fields can be counterproductive, due to the relatively slow adjustment of initial atmospheric fields.

Improved SST fields provide often, but not always (and not always significantly) enhanced forecast performances with respect to the corresponding simulations with native SST fields. Especially in case study 1, the effects close to the Corigliano rain gauge seem to be somehow linked to generally chaotic behaviour. Such as discussed in the case of improved initial

conditions (i.e., the 3DVAR scheme), these outcomes are related to the fact that other sources of uncertainties rather than SST representation hinder enhanced forecast skills. Nevertheless, using more realistic SST fields leads to enough clear changes in the simulation of the atmospheric boundary layer dynamics in both case studies, especially with respect to the configurations with clearly unrealistic fields (i.e., IFS lower boundary conditions). Specifically, in the summer (shorter, convective, highly localized) event higher SST values along the coastlines accelerate flow dynamics, moving faster humid

air towards the coast and moving up precipitation (thus agreeing with the results achieved by Stocchi and Davolio, 2017). On the other hand, in the autumn (longer, caused by a frontal system, widespread) event the higher energy supplied to the system by a continuously warmer sea surface leads to a generalized increase of precipitation amount that, however, does not change substantially neither the spatial pattern nor the timing of the event. The missed change in timing is most probably due to the fact that the stability of the atmospheric boundary layer and the related flow dynamics in case study 2 depend more on

large scale (synoptic) conditions than local factors (that possibly is the same reason why, on the other hand, the 3DVAR scheme is capable to influence more case study 2 than case study 1). Such large-scale conditions are capable to lead to much stronger winds than case study 1 (that is evident comparing Figs. 8 and 16).

Exploring in detail the hydrological impact of case study 2, the analysis must be related to the resolution of the small-scale catchments, where the experiments show that results achieved on larger scales (i.e., at the resolution of the warning areas)



can be 'doubly' reversed. For example, both bias analyses and Taylor diagrams related to the Ancinale River Basin (Figs. 17 and 20) highlight better QPF performances for GFS-based simulations, not so obvious (or even not found) in the analysis of simulations skills on a larger scale. Nevertheless, IFS-based hydrographs are better correlated with those calculated with observed rainfall and peak flow times are closer to observed (it is worth to recall that a quantitative discharge analysis is less

significant in this case, given that only water level observations are available). Contrary to what was found by Yucel et al. (2015), streamflow simulations are not particularly improved by initial data assimilation. Most probably, this result is due to the relatively long forecast periods (from 72 to 96 hours). Indeed, in the 3-day forecast the benefits of the improved initial conditions partially come to light in both catchments even though, interestingly, the best simulation (even if only slightly) is IFS-DA-O, i.e. that using the 3DVAR scheme but not the Medspiration SST fields, meaning that at the high resolution scale

of the catchments analysed there are so many sources of uncertainty that the added value provided by improved SST fields is hidden.

Summarizing, the results achieved in this study show that none of the different versions of the forecasting chain adopted is capable to achieve in all the analysed cases quantitative precipitation and (consequently) streamflow forecast, yet several interesting clues are provided. Specifically, similar to past studies it is shown that the improved representation of SST fields

can significantly change the simulation of the atmospheric boundary layer processes, modifying flow dynamics and/or the amount of precipitated water. Nevertheless, the potentially positive impact of improved SST fields can be easily hidden by several other sources of uncertainty (mainly, the relevance of the choice of the GCM providing boundary conditions). Further improvements in both GCMs (e.g., the higher-resolution IFS cycle since March 2016) and RCMs will reduce uncertainties highlighting more clearly the need of improved SST representation in regional modelling. Emerging

approaches like regional-scale fully-coupled ocean-atmospheric (e.g., within the Baltic Sea Experiment – BALTEX, Gustafsson et al., 1998; Pullen et al., 2003; Ren et al., 2004; Loglisci et al., 2004; Ricchi et al., 2019; Lewis et al., 2019) or ocean-atmospheric-hydrologic (Ruti et al.; 2016, Somot et al., 2018) modelling aim to directly calculate SST fields dynamics. Meanwhile, with the current generation of operational models, a reasonable (yet computationally demanding) solution is to adequately take into account the uncertainty of SST in forecasting chains adopting ensemble approaches also

for this variable.

**Data availability**.    Rainfall data are provided, upon request, by the "Centro Funzionale Multirischi – ARPACAL" (http://www.cfd.calabria.it/). Radar data are provided, upon request, by the Italian National Civil Protection "Centro Funzionale Centrale Rischio Meteo-idrogeologico e Idraulico" (http://www.protezionecivile.gov.it/home). Instruction for

acquiring Medspiration L4 Ultra-High Resolution SSTfnd data are provided at http://cersat.ifremer.fr/thematic-portals/projects/medspiration. Observations used to perform data assimilation are available at https://rda.ucar.edu/. Simulation data are available from the corresponding author upon request.

**Author contribution.** All authors contributed equally to the manuscript.



**Competing interests.** The authors declare that they have no conflict of interest.

**Aknowledgements.** We thank the "Centro Funzionale Multirischi" of the Calabrian Regional Agency for the Protection of
5 the Environment, for providing the observed precipitation data and the Italian National Civil Protection "Centro
Funzionale Centrale Rischio Meteo-idrogeologico e Idraulico" for providing radar data. L. Furnari acknowledges support
from the Programme "POR Calabria FSE/FESR 2014/2020 – Mobilità internazionale di Dottorandi e Assegnisti di
ricerca/Ricercatori di Tipo A" Actions 10.5.6 and 10.5.12."

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





# Figures

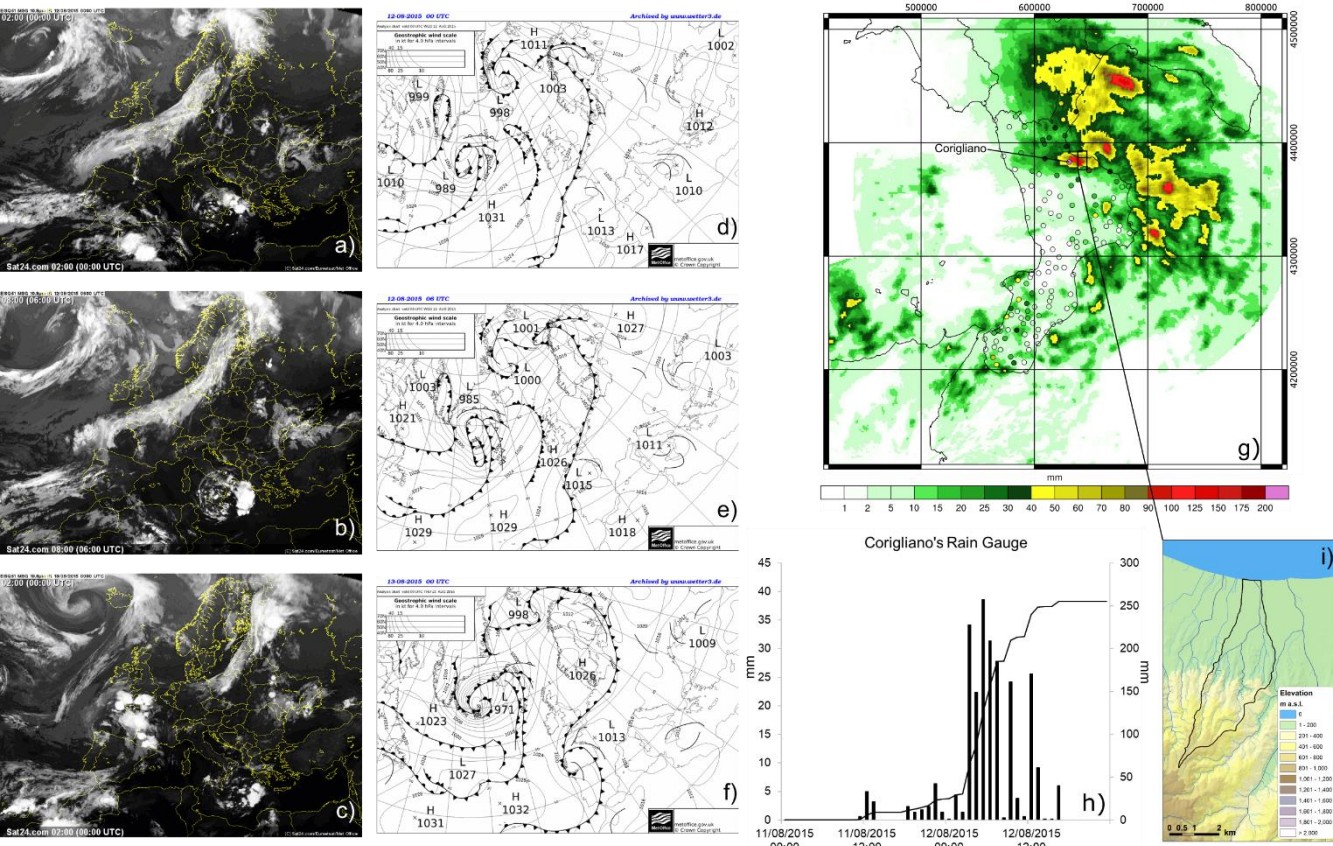

**Fig. 1. From a) to c):** satellite images of the thermal infrared channel (10.8 μm) at: a) 00 UTC 12 August 2015; b) 06 UTC 12 August 2015; c) 00 UTC 13 August 2015, source: www.sat24.com, ©Eumetsat; from d) to f): surface pressure and weather fronts at: d) 00 UTC 12 august 2015; e) 06 UTC 12 august 2015; f) 00 UTC 13 august 2015; source: www.wetter3.de, @Metoffice; g) cumulative rain (mm) observed between 18 UTC 11 August 2015 and 18 UTC 12 August 2015, points represent the weather stations while spatially distributed values represent the radar estimation; h) cumulative and hourly rainfall (mm) observed at the Corigliano rain gauge; i) the Citrea Creek catchment.





**Fig. 2. From a) to c): surface pressure and weather fronts at: a) 00 UTC 12 august 2015; b) 06 UTC 12 august 2015; c) 00 UTC 13 august 2015; source www.wetter3.de, © Metoffice; from d) to g): daily rainfall (mm) observed from 30 October 2015 to 2 November 2015; points represent the weather stations while spatially distributed values represent the radar estimation; h) Ancinale River catchment; i) Bonamico Creek catchment.**





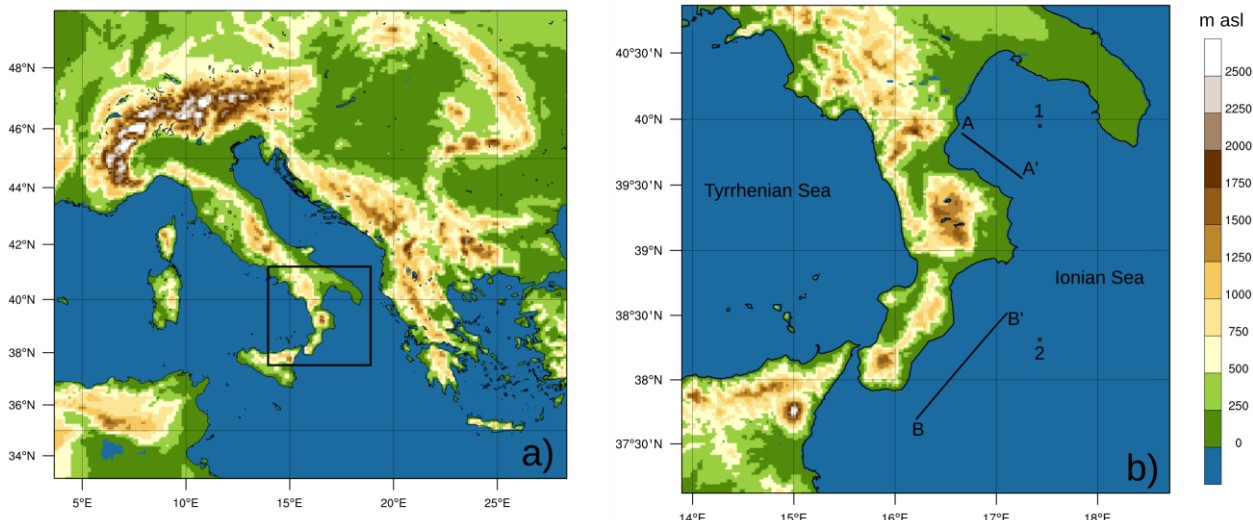

**Fig. 3. a) outer domain with spatial resolution of 10 km; b) inner domain with spatial resolution of 2 km. Points 1 and 2 are considered for evaluating SST evolution locally during the events according to different configurations (Fig. 5 and Fig. 11, respectively). Across sections A-A' and B-B' vertically integrated water vapour fluxes are calculated (Fig. 10 and Fig. 14, respectively).**

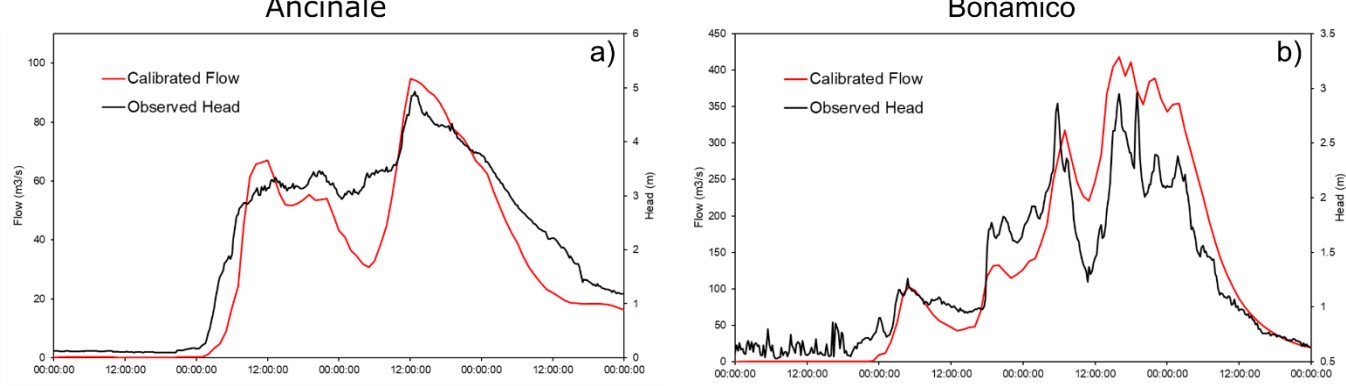

**Fig. 4. Comparison between observed hydrometric levels (m) and calibrated simulated flow (m s⁻³) at: a) Razzona gauging station (Ancinale River), and b) Casignana gauging station (Bonamico Creek).**





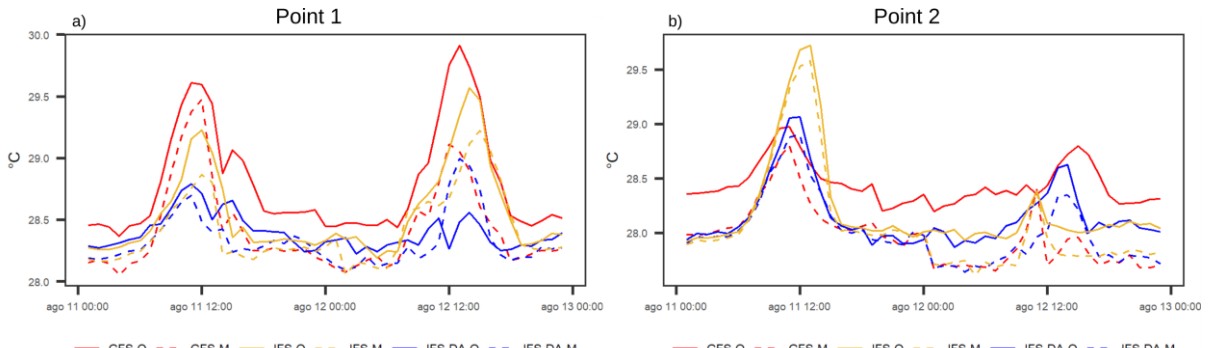

**Fig. 5. Case study 1: time evolution of SSTSK at points 1 and 2 highlighted in fig 3b.**



**Fig. 6. Accumulated precipitation (mm) from 18 UTC 11 August 2015 to 18 UTC 12 August 2015, simulated with different configurations. Observations are superimposed (coloured dots). The small blue (Figs. 6a-b) or white (Figs. 6c-f) stars highlight the accumulated rainfall peaks near Corigliano analysed in detail in Fig. 7.**





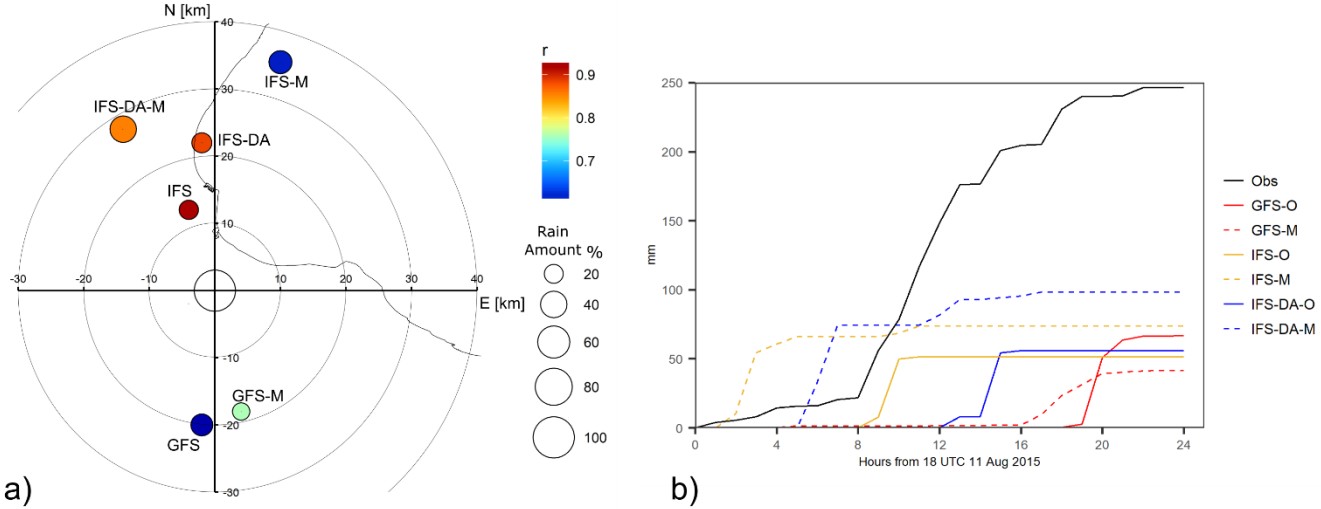

**Fig. 7. Circles in a) are located at the peaks highlighted in Fig. 6 for each of the different configurations; the colours indicate the time correlation, while and the size refers to the percentage rain amount with respect to Corigliano observations; b) temporal accumulated rainfall (mm) observed at Corigliano and simulated by the different peaks.**

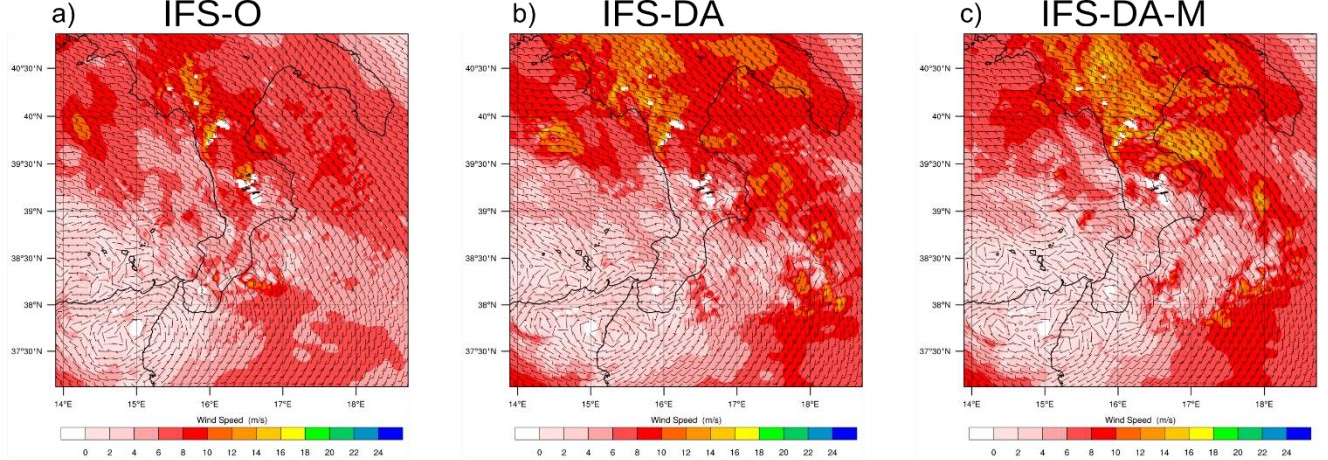

**Fig. 8. Wind direction (barbs) and wind speed (m s⁻¹), shown both as barbs and colour, at 850 hPa at 00 UTC 12 August 2015, for a) IFS-O, b) IFS-DA and c) IFS-DA-M, respectively.**





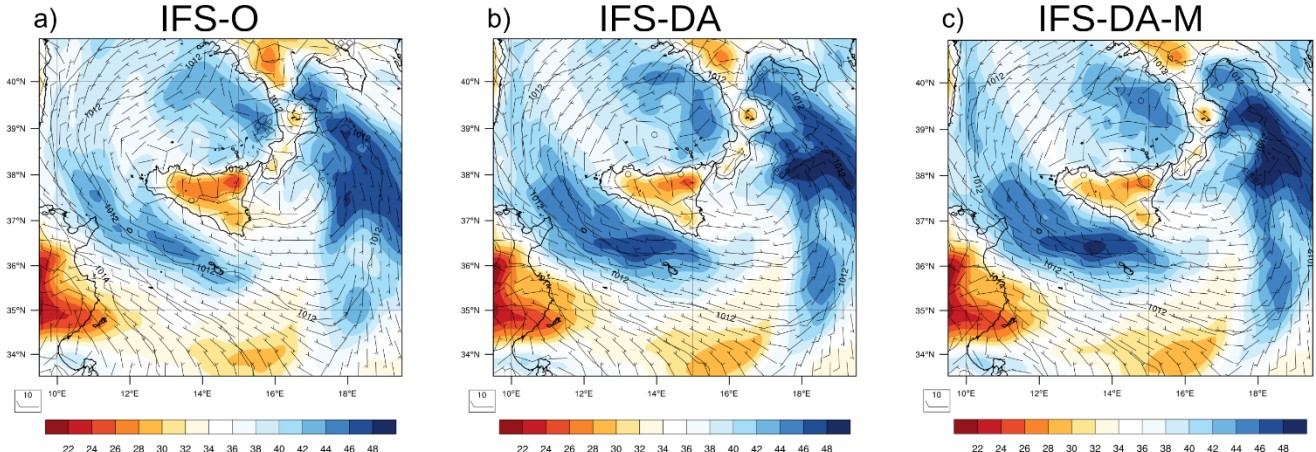

**Fig. 9. Column IWV (kg/kg, colour), sea level pressure (hPa; contours), wind direction and speed (barbs) at 10m height at 03 UTC 12 August 2015 for a) IFS-O, b) IFS-DA and c) IFS-DA-M, respectively.**

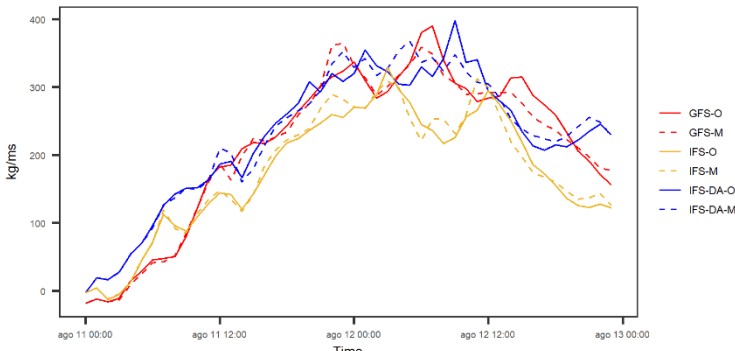

5    **Fig. 10. Time evolution from 00 UTC 11 August 2015 to 00 UTC 13 August 2015 of the vertically integrated water vapour flux (kg m$^{-1}$ s$^{-1}$) crossing section A-A' shown in Fig. 2.**

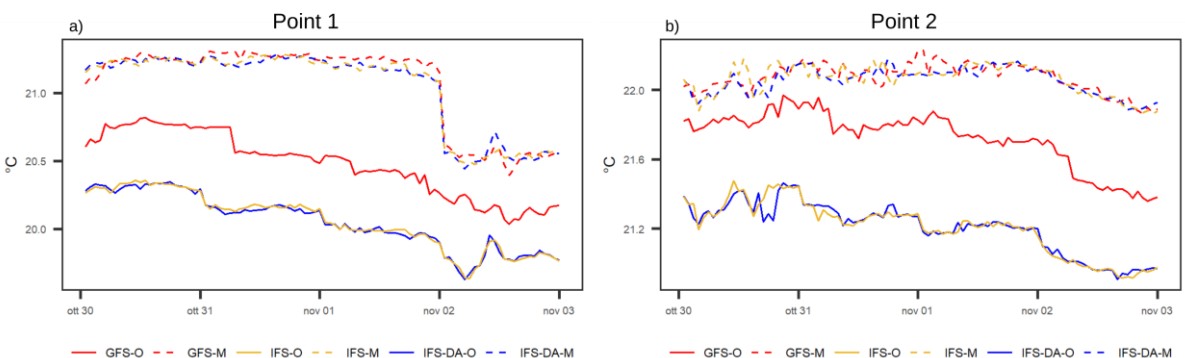

**Fig. 11. Case study 2: time evolution of SSTSK at points 1 and 2 highlighted in fig 3b.**





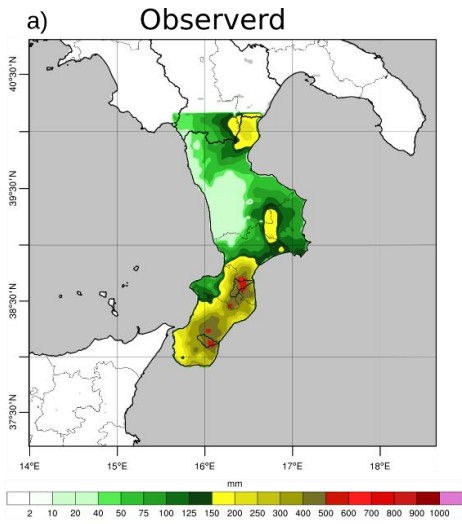

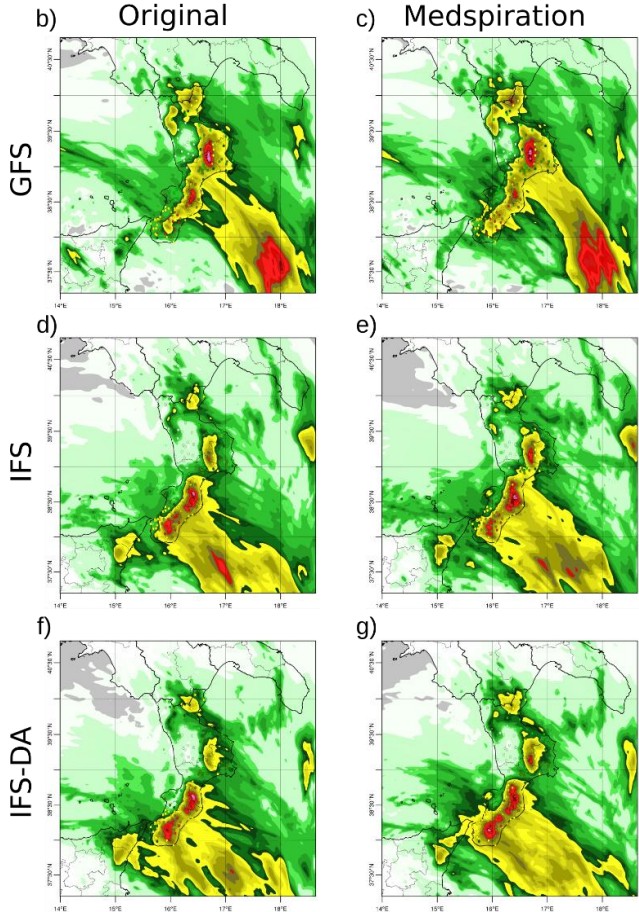

**Fig. 12. Accumulated precipitation (mm) over the whole period of 96 hours starting from 00 UTC 30 October 2015: a) merging of ground measurements to radar observations (refer to text for details); b-g) simulated fields with different configurations. Observations are superimposed (coloured dots).**

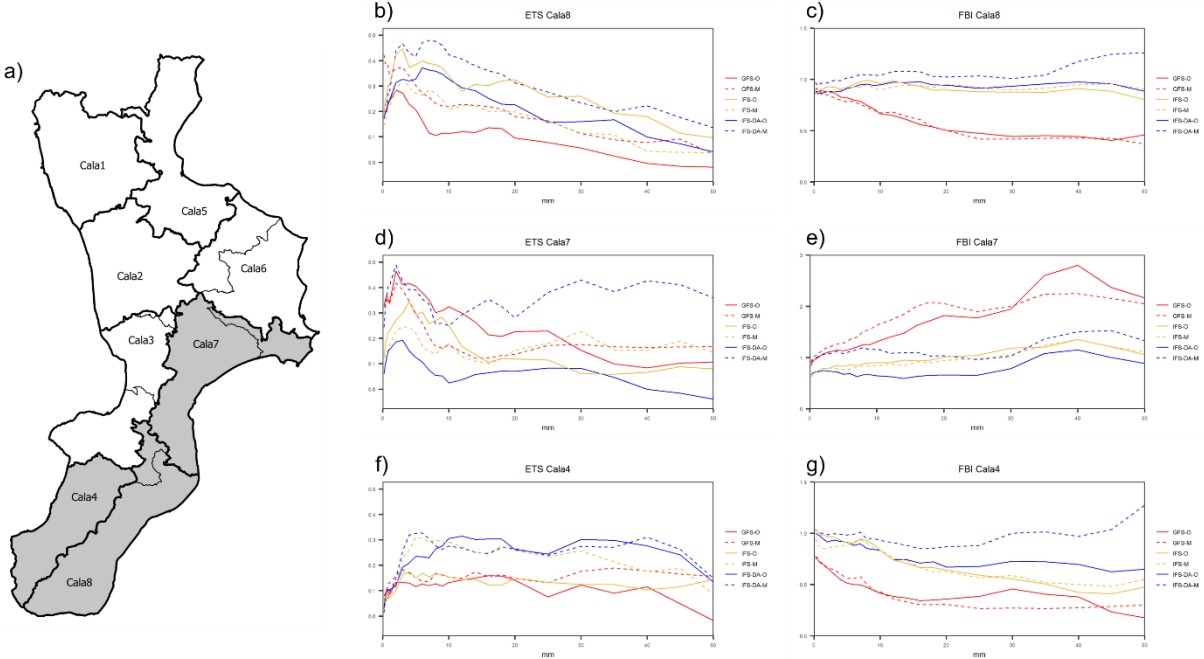

**Fig. 13. b-g)** Categorical scores ETS (Equitable Threat Score) and FBI (Frequency Bias Index) calculated on the rain gauges located in the three Civil Protection warning areas more affected by the event of case study 2, highlighted in a) as grey areas.

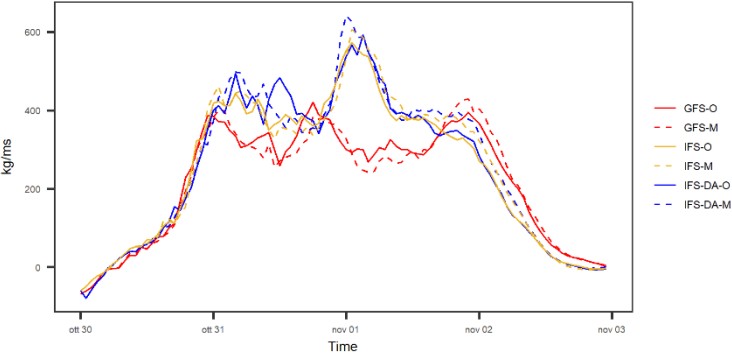

5  **Fig. 14.** Time evolution from 00 UTC 30 October 2015 to 00 UTC 3 November 2015 of the vertically integrated water vapour flux (kg m$^{-1}$ s$^{-1}$) crossing section B-B' shown in Fig. 2.





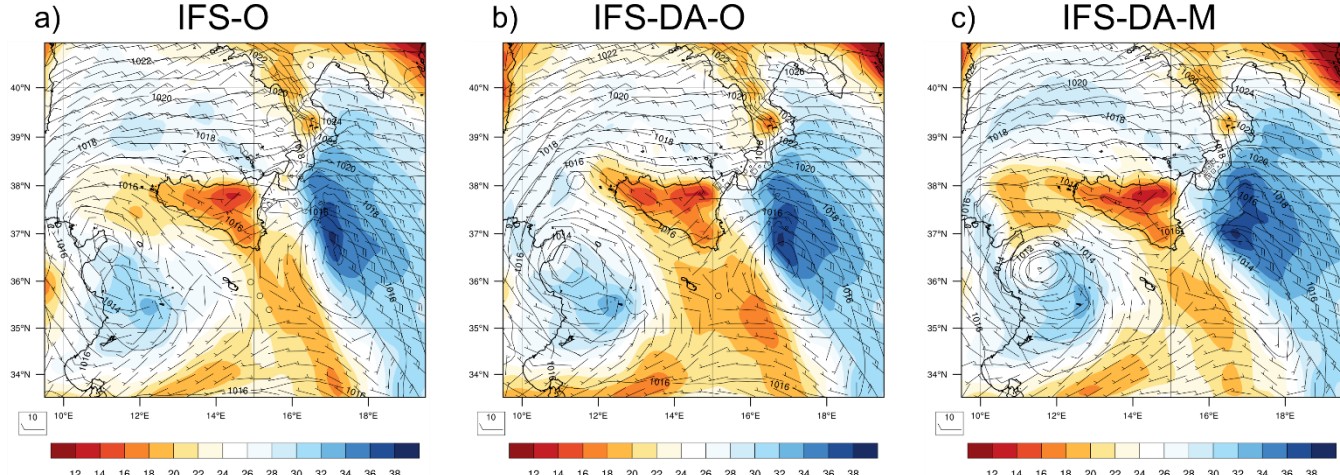

**Fig. 15. Same as Fig. 9, at 21 UTC 31 October 2015.**

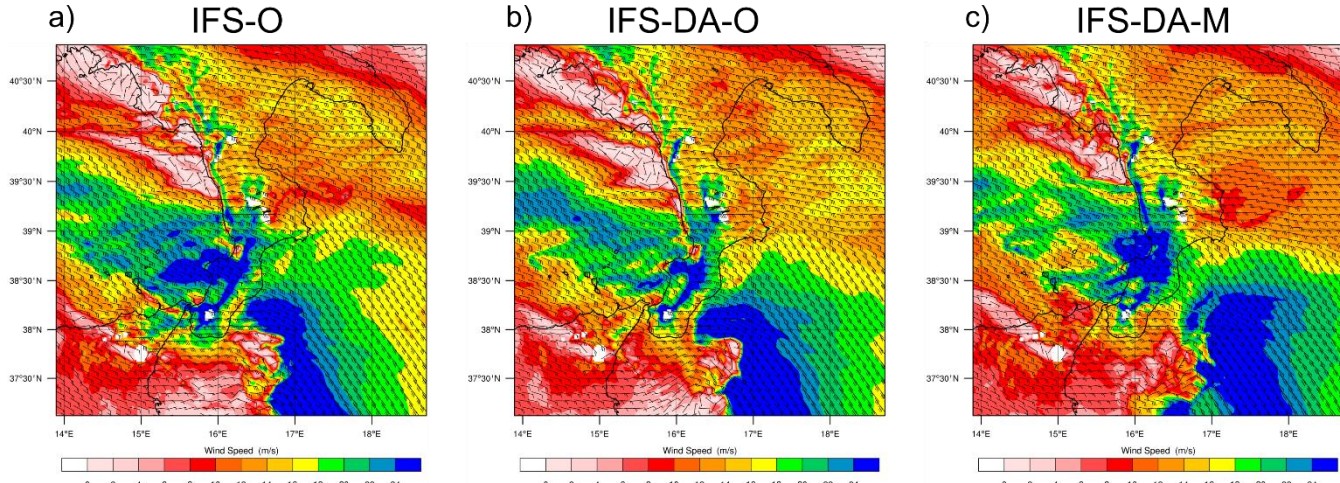

**Fig. 16. Same as Fig. 8, at 21 UTC 31 October 2015.**



**Fig. 17. a) Taylor diagram related to the hourly precipitation series over the Ancinale River catchment simulated by the different configurations forecasting 96 hours, compared to observations; b) resulting hydrographs (m s⁻³) obtained by the different WRF-Hydro simulations compared to observations; c) same as a), but for the Bonamico Creek catchment; d) same as b), but for the Bonamico Creek catchment.**


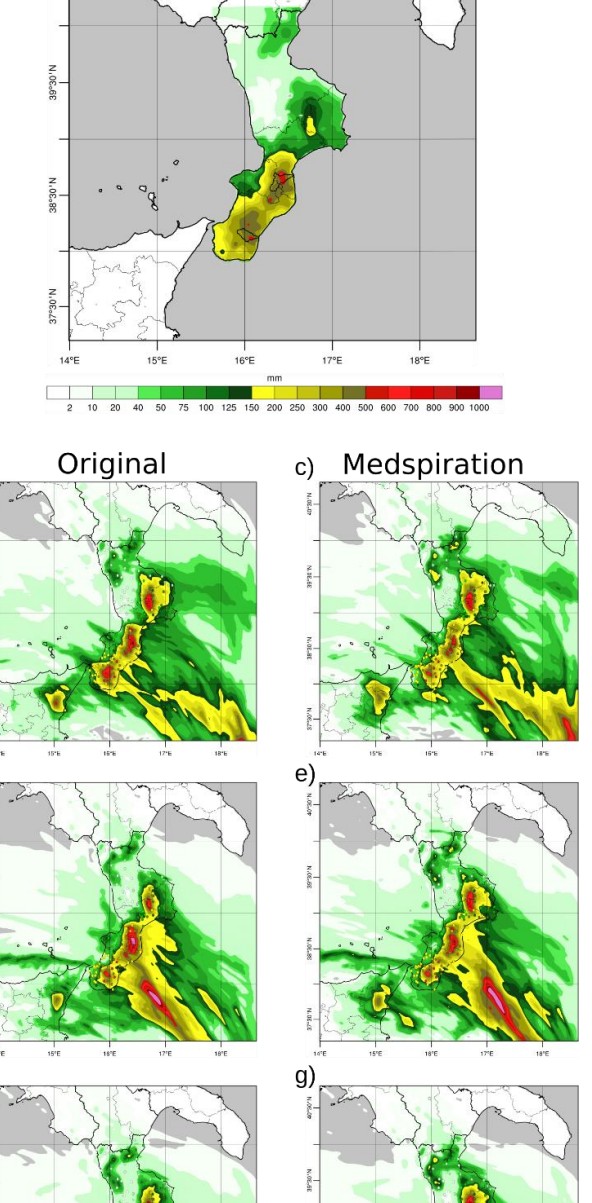

**Fig. 18. Same as Fig.12, but for a 72-hour simulation period, starting from 00 UTC 31 October 2015.**





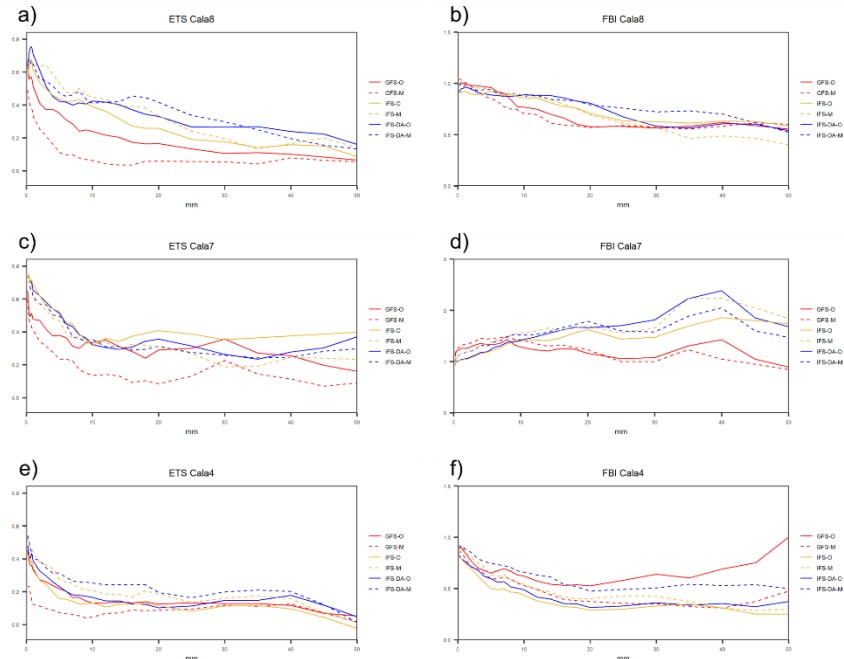

**Fig. 19. Same as Fig. 13, but for a 72-hour simulation period, starting from 00 UTC 31 October 2015.**





**Fig. 20. a) Taylor diagram related to the hourly precipitation series over the Ancinale River catchment simulated by the different configurations forecasting 72 hours, compared to observations; b) resulting hydrographs (m s⁻³) obtained by the different WRF-Hydro simulations compared to observations; c) same as a), but for the Bonamico Creek catchment; d) same as b), but for the Bonamico Creek catchment.**





**TABLES**

**Table 1. Main WRF physical options selected for the study.**

| Component | Scheme Adopted |
|---|---|
| Microphisics Scheme | 2 - Lin-Purdue (Chen and Sun , 2002) |
| PBL Scheme | 2 - MJY (Mellor and Yamada,1982) |
| Shortwawe Radiation Physics Scheme | 1 - Dudhia (Dudhia, 1989) |
| Longwawe Radiation Physics Scheme | 1 - RTTM (Mlawler et al. 1997) |
| Land Surface Model | 2 - Unified NOAH (Tewari et al., 2004) |
| Surface Layer | 2 - Eta Similarity (Janjic, 1994) |
| Cumulus Physics Scheme | 1 - Kain-Fritsch (only D01) (Kain, 2004) |

**Table 2. List of simulations with related acronyms.**

| ID | GCM | SST Source |
|---|---|---|
| GFS-O | GFS 0.25° Forecast | Original |
| GFS-M | | Medspiration |
| IFS-O | IFS-ECMWF Forecast | Original |
| IFS-M | | Medspiration |
| IFS-DA-O | IFS-ECMWF Forecast | Original |
| IFS-DA-M | 3DVAR Assimilation | Medspiration |
| Case Study II | | |
| GFS-O | GFS 0.25° Forecast | Original |
| GFS-M | | Medspiration |
| IFS-O | IFS-ECMWF Forecast | Original |
| IFS-M | | Medspiration |
| IFS-DA-O | IFS-ECMWF Forecast | Original |
| IFS-DA-M | 3DVAR Assimilation | Medspiration |



**Table 3. Calibrated parameters of the off-line WRF-Hydro model for the Ancinale River and the Bonamico Creek.**

| Parameter | Ancinale | Bonamico |
|---|---|---|
| REFKDT | 0.7 | 0.4 |
| SLOPE | 0.30 | 0.30 |
| Z1 (m) | 0.2 | 0.1 |
| Z2 (m) | 0.5 | 0.2 |
| Z3 (m) | 1.20 | 0.5 |
| Z4 (m) | 2 | 0.9 |
| OVROUGHRTFAC | 50 | 50 |
| RETDPRTFAC (mm) | 0 | 15 |
| Manning's roughness coefficient 1st order | 0.1 | 0.1 |
| Manning's roughness coefficient 2nd order | 0.062 | 0.063 |
| Manning's roughness coefficient 3rd order | 0.048 | 0.045 |
| Manning's roughness coefficient 4rt order | 0.033 | 0.031 |





**Table 4. Synoptic table summarizing the main findings for the different case studies. Acronyms: Medsp: Medspiration BC: boundary conditions. CP: Civil Protection.**

| | Case Study 1 | Case Study 2 | |
| --- | --- | --- | --- |
| | 11-12 Aug (48h) | 30 Oct – 2 Nov (96h) | 31 Oct – 2 Nov (72h) |
| Skin SST fields | Generally small differences (slightly higher values with GFS and lower with Medsp), but strong IFS underestimation along coastlines. | Strong IFS underestimation along coastlines. Average Medsp values higher than GFS (about from 0.6 to 0.8 K) and IFS (> 0.8 K, even not considering IFS underestimation along coastlines). Also, with GFS underestimation along coastlines, but overestimation off the Tyrrhenian Sea. | |
| Precipitation amount and spatial pattern | Average rainfall increases in domain D02 moving from IFS to GFS to IFS-DA. GFS rainfall centred to the south-east, IFS and DA show more elongated shapes in the south-north direction. Medsp effect is minor with respect to varying GCM or including DA. | More rainfall in D02 with GFS. Moving from GFS to IFS to IFS-DA, a shift of the biggest rainfall cluster over the sea is observed from north-east to south-west direction. Medsp fields increase average precipitations (about 10%), but do not affect spatial patterns significantly. | GFS-based simulations closer to the IFS-based. Medsp fields increase average precipitations (about 10%), but do not affect spatial patterns significantly. |
| Precipitation timing and scores | Close to the Corigliano rain gauge, GFS-based simulations delay the event. Ingesting Medsp fields accelerate flow dynamics, especially in IFS-based simulations. | Globally, better performances with IFS-DA-M in the 3 CP warning areas analysed. Relevant over- or underforecasts with GFS-based simulations. Medsp fields especially useful to improve 3DVAR scheme, but do not change the timing of the event. | Scores of GFS-based simulations still worse, even with Medsp fields. Also, for IFS-based simulations, Medsp effect is less relevant and not always positive. Substantial over- or underforecasts with almost all simulations. |
| Hydrological impact | Not feasible, because no simulation can forecast reliable precipitation values for the Citrea catchment. | QPF analysis at the catchment scale: for Ancinale River, overforecast with IFS-based simulations, GFS-based much better; for Bonamico Creek, GFS-based underforecast, IFS-based better.<br>Hydrographs, Ancinale: most IFS-based simulations are reasonably correlated with observations (but discharge overforecast), peak flow times close to observed; in GFS-based, volumes closer to observations but peak flow times are not well forecasted.<br>Hydrographs, Bonamico: all hydrographs are not well | QPF analysis at the catchment scale: for Ancinale River, GFS-M is the best simulation, while IFS-based simulations overforecasts are reduced; for Bonamico Creek, GFS-based forecasts bias is reduced, but overall better performances with IFS-based (Taylor diagram).<br>Hydrographs, Ancinale: results similar to 4-day forecasts, best performances with IFS-DA-O.<br>Hydrographs, Bonamico: IFS-based simulations well correlated, peak flow times well forecasted (especially IFS-DA-O). GFS-based simulations poorly correlated, early forecast of the |



| | correlated with observations, peak flow times not well forecasted. | peak flow. |
| --- | --- | --- |