# Peer review of "Impact of high-resolution Sea Surface Temperature representation on the forecast of small Mediterranean catchments hydrological response to heavy precipitation"

_Hydrology and Earth System Sciences, 2019_

## Referee Comment (RC1) · Anonymous Referee #1 · 22 Jul 2019

Review of manuscript "Impact of improved Sea Surface Temperature representation on the forecast of small Mediterranean catchments hydrological response to heavy precipitation", Southern Italy" (hess-2019-345) by Senatore et al.

The manuscript is well written and clearly shows the potential benefit of improved SST information for precipitation and streamflow forecast with WRF and WRF-Hydro. It should therefore be considered for publication in nhess.

Minor comments page 1: It should be mentioned in the abstract that the dynamical

downscaling of the global forecasts are done with the WRF model.

Page 12, line 20-30: it could be mentioned that the hydrographs discussed here are obtained with the offline WRF-Hydro forced with the WRF downscaled forecasts.

Figures Fig. 1: the caption in the elevation legend of panel i) is very small. Couldn't the figure be reorganized so that the size of panel i) looks similar to the size of panels h) and i) in Fig. 2 ?

Figs. 8 and 16. The caption size should be increased. Also, the barbs density should be reduced and the barbs size increased.

Figs. 13 and 19. Caption size should also be increased here.

---

## Referee Comment (RC2) · Anonymous Referee #2 · 10 Sep 2019

Review of: Impact of improved Sea Surface Temperature representation on the forecast of small Mediterranean catchments hydrological response to heavy precipitation (Alfonso Senatore1, Luca Furnari1, Giuseppe Mendicino1)

General comments

============

This paper addresses the sensitivity of precipitation, and thereby river flow, prediction in WRF-Hydro to the choice of SST boundary forcing, with specific focus on 2 case study

periods of heavy precipitation in southern Italy. The work is of general interest, particularly given the increasing effort on developing more integrated hydro-meteorological prediction systems. Overall the paper is well written and contains some original insights. The Introduction is well considered and clear in setting the context for this work, and I found Table 4 to be a very useful and well considered summary, along with most of the discussion presented in Section 5. In particular, the exploration of sensitivity of hydrological application to the regional atmosphere configuration is an important link between regional atmosphere and hydrological communities.

At times I found the presentation difficult to follow, particularly given the large number (6) of different experiments considered and effectively 3 different case study periods discussed. I expand on some specific examples below. My recommendation would therefore be for the authors to aim to undertake a thorough review of the materials and discussions, with a view to reducing the length of the paper, and number of figures, in order to provide a more concise and impactful treatment of the material. I also raise some more specific but equally fundamental concerns below which should be addressed. I encourage the authors to resubmit to HESS after these major revisions.

Specific comments

\_\_\_\_\_

A) There are a large number of figures presented, and some (e.g. Fig 1, Fig 2) have a number of disparate sub-plots, which was more overwhelming than adding clarity. I encourage the authors to reduce the number of figures and sub-plots in general. As an example, the synoptic situation could be more usefully highlighted through Fig 1(b) and (e) only, and in Fig. 2(b) only perhaps.

B) Similarly, the other materials of Fig. 1 might be better placed alongside relevant comparison, such as in Fig. 4 (gauge rainfall comparing with gauged flows), and Fig. 6 (maps of precip accumulation).

C) P5, I30: Consider omitting this paragraph as not relevant to the paper. Perhaps instead it is worth highlighting the potential for higher temporal frequency updating products in the discussion section, alongside highlighting the potential for use of dynamically coupled ocean-atmosphere systems in place of the global-scale boundary conditions. A relevant recent reference here includes the work of: https://www.ocean-sci.net/15/761/2019/os-15-761-2019.html

D) Section 2.2.2: Calibration methodology. I find it strange that a calibration methodology is applied using data from the period of interest, rather than using some external data outside either case study 1 or case study 2 periods. This approach should be defended more clearly, or replaced with a more considered approach. Further, it is not clear which model data were used in the calibration – were all experiments based on the same calibration data, and thereby favouring one of the experiments over others presumably? Given that the study aims to assess the sensitivity of simulations to different configurations, this does not seem a valid approach here. The authors might clarify what effect the calibration had on the predicted flows, or perhaps simply be consistent and use uncalibrated settings for both case study 1 and case study 2.

E) Improved SST: The paper discusses the impact of "different accuracy levels of SST", and makes reference to "improved SST" in several places (e.g. title, abstract, section 3.1 from p8,I5, section 3.2 from p10,I6, Section 4, p15). However, the paper is critically lacking in Fig 4 and Fig 11 for example from lack of SST observations against which to conclude which systems "overestimate", "underestimate" or have "improved" SST. At best, it seems that the authors can illustrate how much the various SST products differ, but a more conclusive treatment might be possible through comparison with insitu observations (e.g. as provided through CMEMS). The authors should review the language of the argument here and throughout, but ideally provide some comparison to available in-situ SST observations.

F) Case Study 1: p9, l24 – I have some issue with the conclusions to Case Study 1, and the lack of treatment of hydrological flows. Given the original nature of going from SST,

СЗ

to precip, to river flows within the paper, this appears to be an omission, even if the result is that there is no sensitivity in this case. Without the full 'end-to-end' treatment here, it seems confusing to justify the exploration of both Case Study 1 and Case Study 2, such that for brevity I wonder what is lost from the paper by only concentrating on Case Study 2 period. This seems to be supported by more limited assessment (e.g. no categorical statistic assessment) for Case Study 1 period. The authors should clarify this final paragraph, and better justify the need for treatment of both period 1 and 2.

G) Case Study 2: I was not clear on the need for separate discussions of both 4-day and 3-day duration simulations, and again this added to the length and more confused presentation. To what extent are the authors conclusions altered by focussing only on discussion of 3-day (or 4-day) results for example? The authors should better justify the treatment of both lead times, and preferably condense the material into a more concise manuscript, perhaps highlighting the key messages of impact of lead time in the discussion. There are opportunities to consolidate some of this information within the Figures, e.g. by presenting the Taylor diagram summaries of 3-day and 4-day results alongside each other (using filled/unfilled symbols to represent different lead times perhaps).

H) It would be useful to highlight some discussion of the relative sensitivity to precipitation to SST shown here (e.g. Fig 6, Fig 12) relative to say the typical ensemble spread expected through other approaches to represent simulation uncertainty.

I) Figure 4 – given above discussion about calibration approach, please include uncalibrated flow results here. Please also clarify the difference between Fig 4 and Fig 17 b) and d). This is another example where a more concise and consolidated presentation of results would improve the paper. For example, which experiment does the "calibrated flow" in Fig 4 show, and why are results from all experiments not included here to provide a more immediately conclusive discussion?

J) Figure 6 – it is hard to distinguish the gauge site information in the shaded circles.

Consider making these larger, perhaps as unshaded circle outlines, or providing in a separate sub-plot of gauge-only observations. Again, some overlap of information with Fig 1g here.

K) Figure 8 – consider removing, given information contained in Fig 9.

L) Figure 12 – it is confusing to have different treatments of the observed precip for different cases and across different figures, e.g. radar and gauge plots in Fig 1 and Fig 2 (quite hard to see gauge sites in small panels), gauge points only in Fig 6, merged gauge/radar in Fig 12a. Authors should aim for more consistent treatment, perhaps across fewer figures, to provide a clearer discussion.

M) Figure 16 – could this be omitted (or provided in Supplementary) without any loss of understanding?

N) Table 2 – this seems to have repeated information between the top section and 'case study 2'. I would just leave the top part for brevity and clarity.

Technical corrections

\_\_\_\_\_

a) While overall the paper is well written, there are occasional places where some improvement to the English style would improve readability. As a specific example, the first sentence of Section 2.1 should be reviewed. Others might be picked up through further edit by the authors and again in the paper production process.

b) Figure 2 - caption has incorrect dates and time for this case I think.

c) P2, I5-6: Please provide specific references to the initiatives listed.

d) P5, I22: Useful to translate 0.25deg. GFs resolution into an approximate grid scale length for region of interest here (i.e. how does this compare to the 16 km IFS?).

e) P8, I5: I did not follow this point - consider rewording

f) Figure 12 - typo in 'observed' on panel a) ć P14, l6: Consider rewording

---

## Author Comment (AC1) · 7 Oct 2019

**We warmly thank the referee for his/her thoroughly review of the paper and positive feedback. Please find below our answers to the items raised (bold text).**

Page 1: It should be mentioned in the abstract that the dynamical downscaling of the global forecasts are done with the WRF model.

**We already indicated that we used the "atmosphere-hydrology modelling system WRF-Hydro in its uncoupled version", but we will strive to be clearer in the**

[Figure]

**revised version of the manuscript, following the reviewer's suggestion.**

Page 12, line 20-30: it could be mentioned that the hydrographs discussed here are obtained with the offline WRF-Hydro forced with the WRF downscaled forecasts.

**Also in this case, we will strive to be more clear and specific in the revised version of the manuscript.**

Figures Fig. 1: the caption in the elevation legend of panel i) is very small. Couldn't the figure be reorganized so that the size of panel i) looks similar to the size of panels h) and i) in Fig. 2 ?

Figs. 8 and 16. The caption size should be increased. Also, the barbs density should be reduced and the barbs size increased.

Figs. 13 and 19. Caption size should also be increased here.

**Thank you for these suggestions. Following comments of both Referee #1 and #2, all the figures will be checked and reviewed.**

―――――――――――――――

---

## Author Comment (AC2) · 7 Oct 2019

We warmly thank the referee for his/her comments and thorough review of our paper, aimed at improving its quality and making it more effective. Please find below our answers to all items raised (bold text).

General comments

\_\_\_\_\_
This paper addresses the sensitivity of precipitation, and thereby river flow, prediction in WRF-Hydro to the choice of SST boundary forcing, with specific focus on 2 case study periods of heavy precipitation in southern Italy. The work is of general interest, particularly given the increasing effort on developing more integrated hydro-meteorological prediction systems. Overall the paper is well written and contains some original insights. The Introduction is well considered and clear in setting the context for this work, and I found Table 4 to be a very useful and well considered summary, along with most of the discussion presented in Section 5. In particular, the exploration of sensitivity of hydrological application to the regional atmosphere configuration is an important link between regional atmosphere and hydrological communities. At times I found the presentation difficult to follow, particularly given the large number (6) of different experiments considered and effectively 3 different case study periods discussed. I expand on some specific examples below. My recommendation would therefore be for the authors to aim to undertake a thorough review of the materials and discussions, with a view to reducing the length of the paper, and number of figures, in order to provide a more concise and impactful treatment of the material. I also raise some more specific but equally fundamental concerns below which should be addressed. I encourage the authors to resubmit to HESS after these major revisions.

We thank the referee for the positive feedback and acknowledge the need for making the paper more concise and the message given more impactful. We will strive to reduce the paper length and the number of figures. As explained in more detail below, specifically, we plan to merge the Sections concerning the 4-day and 3-day simulations in Case Study 2, and to reduce the number of figures from 20 to 15, making also less complex the first two multi-panel figures. Responses to specific comments are provided below, including issues not related to the presentation quality.

Specific comments

HESSD
A) There are a large number of figures presented, and some (e.g. Fig 1, Fig 2) have a number of disparate sub-plots, which was more overwhelming than adding clarity. I encourage the authors to reduce the number of figures and sub-plots in general. As an example, the synoptic situation could be more usefully highlighted through Fig 1(b) and (e) only, and in Fig. 2(b) only perhaps.

We agree with the referee that reducing the number of figures and subplots would help a lot to make our paper clearer. While details about single figures are explained responding to the relative comments, our general strategy is the following:

- Subplots in Figs. 1 and 2 will be reduced removing several synoptic maps, as suggested by the referee;

- Figs. 8 and 16 will be moved in the Supplement, according to referee's suggestion;

- Figures related to the 4-day and 3-day simulations in Case Study 2 will be revised. Specifically, Figs. 13 an 19 will be merged, such as Figs. 17 and 20, while Fig. 18 will be moved to Supplement.

B) Similarly, the other materials of Fig. 1 might be better placed alongside relevant comparison, such as in Fig. 4 (gauge rainfall comparing with gauged flows), and Fig. 6 (maps of precip accumulation).

In our opinion, the information related to the impact in terms of precipitation on the ground is better placed next to the synoptic maps, to provide a comprehensive overview of the event. We believe that showing rain gauges data overlying radar information offers a quali-quantitative estimate of the impact both on the ground and over the sea. Gauge sites in Figs. 1 and 2 will be better highlighted, enlarging both their symbols and the maps.

Hydrographs in Fig. 4 relate to the event described in Fig. 2, not Fig. 1. However,
we will improve the figure adding the averaged rainfall on the catchments as histograms on the top horizontal axes.

Concerning Fig. 6, it will be improved adding a map of merged precipitation on the land (Sinclair and Pergram, 2005), such as it is in Fig. 12. Though we acknowledge the problem of possible overlap of information with the rainfall map in Fig. 1 (for which we will adopt specific countermeasures, as it is specified in our responses to comments J and L), we believe that this map is useful here, since it facilitates the quantitative comparison between forecasted and observed rain fields (at least on the land), avoiding misinterpretations given by the substantial underestimate of radar.

C) P5, I30: Consider omitting this paragraph as not relevant to the paper. Perhaps instead it is worth highlighting the potential for higher temporal frequency updating products in the discussion section, alongside highlighting the potential for use of dynamically coupled ocean-atmosphere systems in place of the globalscale boundary conditions. A relevant recent reference here includes the work of: https://www.oceansci.net/15/761/2019/os-15-761-2019.html

According to the referee's comment, this paragraph will be removed, while in the discussion section we will focus more on the suggested topics. It is worthwhile to underline that the theme of dynamically coupled ocean-atmosphere systems is already hinted at in the Discussions, providing already also the suggested reference.

D) Section 2.2.2: Calibration methodology. I find it strange that a calibration methodology is applied using data from the period of interest, rather than using some external data outside either case study 1 or case study 2 periods. This approach should be defended more clearly, or replaced with a more considered approach. Further, it is not clear which model data were used in the calibration - were all experiments based on the same calibration data, and thereby favouring one of the experiments over others
presumably? Given that the study aims to assess the sensitivity of simulations to different configurations, this does not seem a valid approach here. The authors might clarify what effect the calibration had on the predicted flows, or perhaps simply be consistent and use uncalibrated settings for both case study 1 and case study 2.

As stated in the Introduction (P3, L9), we focus on the impact of SST on the whole forecasting chain, up to the streamflow, in the context of the uncertainty linked to initial and boundary conditions in regional modelling (different GCMs and DA). With this aim, in our opinion, it is reasonable to minimize the other sources of uncertainty, among which that linked to the hydrological model. The model calibrated with data of the period of interest provides the "best" hydrological modelling result achievable if precipitation over the catchment was perfectly forecasted. Therefore, the more the precipitation forecast worsens, the more the hydrograph differs from the "optimum" (and we can assess how much). Working with a hydrological model calibrated using data from another period would have made this analysis less straightforward. E.g., errors in precipitation forecast (equifinality problem).

It is true that all the experiments were based on the same calibration data, but such data were not derived from any of the experiments, rather from observed rainfall. Hence, none of the experiments was favoured over others. We realize that we were not clear enough, we will strive to explain better our procedure.

Finally, the effect of calibration on the predicted flow will be better highlighted adding uncalibrated flow results to Fig. 4 (such as suggested in the next comment I) and discussing them.

E) Improved SST: The paper discusses the impact of "different accuracy levels of SST", and makes reference to "improved SST" in several places (e.g. title, abstract, section 3.1 from p8,I5, section 3.2 from p10,I6, Section 4, p15). However, the paper is critically
lacking in Fig 4 and Fig 11 for example from lack of SST observations against which to conclude which systems "overestimate", "underestimate" or have "improved" SST. At best, it seems that the authors can illustrate how much the various SST products differ, but a more conclusive treatment might be possible through comparison with insitu observations (e.g. as provided through CMEMS). The authors should review the language of the argument here and throughout, but ideally provide some comparison to available in-situ SST observations.

We agree that, without an explicit comparison with in-situ observations, it is more correct to refer simply to "higher-resolution", rather than "more accurate" or "improved" SST representation, even though in some cases (e.g., compared with the IFS SST fields along coastlines) the improvement is rather obvious. We will review the language used for this topic throughout the manuscript. Nevertheless, we also aim to follow the interesting suggestion about a comparison of the models' skin SST fields with observations. We've made a preliminary search in the CMEMS database, in particular in the CORA database (Cabanes et al., 2013) in the latest version released (5.2, April 2019), looking for suitable observations (i.e., SST measurements at the sea-surface interface) for the dates of interest. We found some useful data, even though rather at the border of the external domain (Fig. 1 below; other data were available more towards the centre of the external domain and even in the innermost domain, but at depths from about 0.3 m to 0.7 m). We intend to provide the results of this analysis even if, for the sake of brevity, in the Supplement.

F) Case Study 1: p9, l24 - I have some issue with the conclusions to Case Study 1, and the lack of treatment of hydrological flows. Given the original nature of going from SST, to precip, to river flows within the paper, this appears to be an omission, even if the result is that there is no sensitivity in this case. Without the full 'end-to-end' treatment here, it seems confusing to justify the exploration of both Case Study 1 and Case Study 2, such that for brevity I wonder what is lost from the paper by only concentrating on
Case Study 2 period. This seems to be supported by more limited assessment (e.g. no categorical statistic assessment) for Case Study 1 period. The authors should clarify this final paragraph, and better justify the need for treatment of both period 1 and 2.

We agree with the referee that, for the sake of completeness, also Case Study 1 needs an "end-to-end" treatment, although there is no sensitivity concerning the hydrological impact in the Citrea Creek catchment. Our study shows that predictability at the scale of small/very small catchments is very limited for this kind of highly convective events, characterized by scattered and chaotic precipitation spatial patterns. We will modify the paragraph and provide details about accumulated precipitation and related peak flow in the Citrea Creek through a specific table.

G) Case Study 2: I was not clear on the need for separate discussions of both 4-day and 3-day duration simulations, and again this added to the length and more confused presentation. To what extent are the authors conclusions altered by focussing only on discussion of 3-day (or 4-day) results for example? The authors should better justify the treatment of both lead times, and preferably condense the material into a more concise manuscript, perhaps highlighting the key messages of impact of lead time in the discussion. There are opportunities to consolidate some of this information within the Figures, e.g. by presenting the Taylor diagram summaries of 3-day and 4-day results alongside each other (using filled/unfilled symbols to represent different lead times perhaps).

We first focused on the 4-day forecast to embrace the event for its entire duration. Then, we wondered about the effects of the relatively long lead time and focused also on 3-day forecasts (which, indeed, provided some differences in the hydrological impact). We realize that the discussion could be made more straightforward. Following the reviewer's suggestion, Sections 3.2.1 and 3.2.2 will be merged and shortened, highlighting the key messages of the lead time impact in the discussion. Furthermore, such as stated before (response to comHESSD
**ment A), Figs. 13 and 19 will be merged, such as Figs. 17 and 20, while Fig. 18 will be moved to Supplement.**

H) It would be useful to highlight some discussion of the relative sensitivity to precipitation to SST shown here (e.g. Fig 6, Fig 12) relative to say the typical ensemble spread expected through other approaches to represent simulation uncertainty.

We thank the referee for this useful suggestion. We will discuss it, referring both to the literature and to some not yet published results, where we downscaled the whole ECMWF ensemble forecast for the August event, finding a level of uncertainty substantially higher than that calculated for these experiments.

I) Figure 4 - given above discussion about calibration approach, please include uncalibrated flow results here. Please also clarify the difference between Fig 4 and Fig 17 b) and d). This is another example where a more concise and consolidated presentation of results would improve the paper. For example, which experiment does the "calibrated flow" in Fig 4 show, and why are results from all experiments not included here to provide a more immediately conclusive discussion?

As stated before (response to comment D), uncalibrated flow results will be included in Fig. 4 and discussed in the text. Furthermore, as explained in the same response, we reiterate here that calibrated data were not derived from any of the experiments, rather from observed meteorological variables. Therefore, the main difference between Fig. 4 and Figs. 17a and 17b is that the former, showing the comparison between observed hydrometric levels and simulated flow calibrated with observed precipitation, highlights how the reference discharge is retrieved for further analyses and its behaviour against available observations; the latter, once such reference discharge is established, highlights how much the forecasted discharges with the modelling chains differ from it. Though we acknowledge that the reference discharge is shown in both figures, we believe that this "double" comparison makes the paper more intelligible, leaving the reader first HESSD
focus on calibration issues, then on the uncertainty of the hydrological output due to the GCM and SST boundary conditions. Therefore, for the sake of clarity, we would like to leave it.

J) Figure 6 - it is hard to distinguish the gauge site information in the shaded circles. Consider making these larger, perhaps as unshaded circle outlines, or providing in a separate sub-plot of gauge-only observations. Again, some overlap of information with Fig 1g here.

We agree that gauge site information here is redundant. The shaded circles will be removed. On the other hand, gauge sites in Figs. 1 will be better highlighted, enlarging both their symbols and the map (as we stated in our response to comment B). Fig. 6 will be improved adding a map of merged precipitation on the land (Sinclair and Pergram, 2005), for facilitating the quantitative comparison between forecasted and observed rain fields on the land (please, refer again to our response to comment B).

K) Figure 8 - consider removing, given information contained in Fig 9.

**We agree with the referee. Figure 8 will be provided in the Supplemental.**

L) Figure 12 - it is confusing to have different treatments of the observed precip for different cases and across different figures, e.g. radar and gauge plots in Fig 1 and Fig 2 (quite hard to see gauge sites in small panels), gauge points only in Fig 6, merged gauge/radar in Fig 12a. Authors should aim for more consistent treatment, perhaps across fewer figures, to provide a clearer discussion.

We agree that maps of observed precipitation need more consistent treatment across the paper. As outlined in the previous answers (particularly to comments B and J), our strategy is the following:

- Figs. 1 and 2 will show rain gauges data overlying radar information. We opt for this solution because we believe, in the context of a general description of

HESSD
the events, that this is the best way to provide both quantitative information of precipitation at specific points on the land and qualitative information about the spatial precipitation patterns all over the domain (in the latter case we refer to "qualitative" information because of the radar underestimate, highlighted by the overlying of rain gauges);

- Figs. 6 and 12 (and Fig. 18 that will be moved to Supplement) will show only distributed rain fields, either simulated by the WRF configurations or reconstructed using the merging procedure proposed by Sinclair and Pergram (2005). This choice is made to favour the quantitative comparison between the amount of rain observed and simulated on the land, exploiting both the quantitative information provided by the monitoring network on the ground and the rainfall pattern description of the radar network.

M) Figure 16 - could this be omitted (or provided in Supplementary) without any loss of understanding?

**Figure 16 will be provided in the Supplemental.**

N) Table 2 - this seems to have repeated information between the top section and 'case study 2'. I would just leave the top part for brevity and clarity.

**We thank the referee for this comment. It was an oversight, we will remove the bottom part of the Table.**

**Technical corrections**

\_\_\_\_\_

a) While overall the paper is well written, there are occasional places where some improvement to the English style would improve readability. As a specific example, the first sentence of Section 2.1 should be reviewed. Others might be picked up through further edit by the authors and again in the paper production process.
**The writing style will be completely re-checked.**

b) Figure 2 - caption has incorrect dates and time for this case I think.

**The referee is right. We will correct the dates and time in the caption.**

c) P2, I5-6: Please provide specific references to the initiatives listed.

**We will do.**

d) P5, I22: Useful to translate 0.25deg. GFs resolution into an approximate grid scale length for region of interest here (i.e. how does this compare to the 16 km IFS?).

**Will do. At these latitudes, 0.25deg is about 27 km.**

e) P8, I5: I did not follow this point - consider rewording

**We will do, in the framework of an overall re-check of the writing style.**

f) Figure 12 - typo in 'observed' on panel a)

**Thank you, we will correct.**

P14, I6: Consider rewording

We will do, in the framework of an overall re-check of the writing style.

**References:**

Cabanes, C., Grouazel, A., von Schuckmann, K., Hamon, M., Turpin, V., Coatanoan, C., Paris, F., Guinehut, S., Boone, C., Ferry, N., de Boyer Montégut, C., Carval, T., Reverdin, G., Pouliquen, S., and Le Traon, P.-Y.: The CORA dataset: validation and diagnostics of in-situ ocean temperature and salinity measurements, Ocean Sci., 9, 1-18, https://doi.org/10.5194/os-9-1-2013, 2013.

Sinclair, S. and Pegram, G.: Combining radar and rain gauge rainfall estimates using conditional merging. Atmosph. Sci. Lett., 6: 19-22. https://doi.org/10.1002/asl.85, 2005. Interactive comment

---

## Author Response (AR1)

**We warmly thank the referees for their comments and thorough review of our paper, aimed at improving its quality and making it more effective. Please find below our answers to the items raised (bold text). All page(s) and line(s) reference relate to the marked-up manuscript version attached at the end of this document.**

Referee #1

Page 1: It should be mentioned in the abstract that the dynamical downscaling of the global forecasts are done with the WRF model.
**We've added the sentence "applying the WRF mesoscale model for the dynamical downscaling" in the Abstract (P1 L18).**

Page 12, line 20-30: it could be mentioned that the hydrographs discussed here are obtained with the offline WRF-Hydro forced with the WRF downscaled forecasts.
**We've added the sentence "performed with the offline WRF-Hydro model forced with the WRF downscaled forecasts" (P14 L11).**

Figures Fig. 1: the caption in the elevation legend of panel i) is very small. Couldn't the figure be reorganized so that the size of panel i) looks similar to the size of panels h) and i) in Fig. 2 ?
Figs. 8 and 16. The caption size should be increased. Also, the barbs density should be reduced and the barbs size increased.
Figs. 13 and 19. Caption size should also be increased here.
**Thank you for these suggestions. Following comments of both Referee #1 and #2, all the figures have been checked and reviewed.**

Referee #2

General comments
================
This paper addresses the sensitivity of precipitation, and thereby river flow, prediction in WRF-Hydro to the choice of SST boundary forcing, with specific focus on 2 case study periods of heavy precipitation in southern Italy. The work is of general interest, particularly given the increasing effort on developing more integrated hydro-meteorological prediction systems. Overall the paper is well written and contains some original insights. The Introduction is well considered and clear in setting the context for this work, and I found Table 4 to be a very useful and well considered summary, along with most of the discussion presented in Section 5. In particular, the exploration of sensitivity of hydrological application to the regional atmosphere configuration is an important link between regional atmosphere and hydrological communities.
At times I found the presentation difficult to follow, particularly given the large number (6) of different experiments considered and effectively 3 different case study periods discussed. I expand on some specific examples below. My recommendation would therefore be for the authors to aim to undertake a thorough review of the materials and discussions, with a view to reducing the length of the paper, and number of figures, in order to provide a more concise and impactful treatment of the material. I also raise some more specific but equally fundamental concerns below which should be addressed. I encourage the authors to resubmit to HESS after these major revisions.

**We thank the referee for the positive feedback and acknowledge the need for making the paper more concise and the message given more impactful. As explained in more detail below, we have merged the Sections concerning the 4-day and 3-day simulations in Case Study 2, and reduced the number of figures from 20 to 15, making also less complex the first two multi-panel figures. Responses to specific comments are provided below, including issues not related to the presentation quality.**

Specific comments
==============
A) There are a large number of figures presented, and some (e.g. Fig 1, Fig 2) have a number of disparate sub-plots, which was more overwhelming than adding clarity. I encourage the authors to reduce the number of figures and sub-plots in general. As an example, the synoptic situation could be more usefully highlighted through Fig 1(b) and (e) only, and in Fig. 2(b) only perhaps.
**We agree with the referee that reducing the number of figures and subplots would help a lot to make our paper clearer. While details about single figures are explained responding to the relative comments, the general strategy that we followed is outlined here:**
**- Subplots in Figs. 1 and 2 have been reduced removing several synoptic maps, as suggested by the referee;**
**- Previous Figs. 8 and 16 have been moved in the Supplement, according to referee's suggestion;**
**- Figures related to the 4-day and 3-day simulations in Case Study 2 have been revised. Specifically, previous Figs. 13 and 19 have been merged (now Fig. 12), such as previous Figs. 17 and 20 (now Fig. 15), while previous Fig. 18 have been moved to Supplement.**

B) Similarly, the other materials of Fig. 1 might be better placed alongside relevant comparison, such as in Fig. 4 (gauge rainfall comparing with gauged flows), and Fig. 6 (maps of precip accumulation).
**In our opinion, the information related to the impact in terms of precipitation on the ground is better placed next to the synoptic maps, to provide a comprehensive overview of the event. We believe that showing rain gauges data overlying radar information offers a quali-quantitative estimate of the impact both on the ground and over the sea. Gauge sites in Figs. 1 and 2 have been better highlighted, enlarging both their symbols and the maps.**
**Hydrographs in Fig. 4 relate to the event described in Fig. 2, not Fig. 1. However, the figure has been improved adding the averaged rainfall on the catchments as histograms on the top horizontal axes.**
**Concerning Fig. 6, it has been improved adding a map of merged precipitation on the land (Sinclair and Pergram, 2005). Though we acknowledge the problem of possible overlap of information with the rainfall map in Fig. 1 (for which we have adopted specific countermeasures, as it is specified in our responses to comments J and L), we believe that this map is useful here, since it facilitates the quantitative comparison between forecasted and observed rain fields (at least on the land), avoiding misinterpretations given by the substantial underestimate of radar.**

C) P5, l30: Consider omitting this paragraph as not relevant to the paper. Perhaps instead it is worth highlighting the potential for higher temporal frequency updating products in the discussion section, alongside highlighting the potential for use of dynamically coupled ocean-atmosphere systems in place of the global-scale boundary conditions. A relevant recent reference here includes the work of: https://www.oceansci.net/15/761/2019/os-15-761-2019.html
**According to the referee's comment, this paragraph has been removed, while in the discussion section we have also focused more on the suggested topics (P18 LL21-25).**

D) Section 2.2.2: Calibration methodology. I find it strange that a calibration methodology is applied using data from the period of interest, rather than using some external data outside either case study 1 or case study 2 periods. This approach should be defended more clearly, or replaced with a more considered approach. Further, it is not clear which model data were used in the calibration – were all experiments based on the same calibration data, and thereby favouring one of the experiments over others presumably? Given that the study aims to assess the sensitivity of simulations to different configurations, this does not seem a valid approach here. The authors might clarify what effect the calibration had on the predicted flows, or perhaps simply be consistent and use uncalibrated settings for both case study 1 and case study 2.
**As stated in the Introduction (P3, L14), we focus on the impact of SST on the whole forecasting chain, up to the streamflow, in the context of the uncertainty linked to initial and boundary conditions in regional modelling (different GCMs and DA). With this aim, in our opinion, it is reasonable to minimize the other sources of uncertainty, among which that**

**linked to the hydrological model. The model calibrated with data of the period of interest provides the "best" hydrological modelling result achievable if precipitation over the catchment was perfectly forecasted. Therefore, the more the precipitation forecast worsens, the more the hydrograph differs from the "optimum" (and we can assess how much). Working with an hydrological model calibrated using data from another period would have made this analysis less straightforward. E.g., errors in precipitation forecast could have been compensated by errors in discharge forecast (equifinality problem).**

**It is true that all the experiments were based on the same calibration data, but such data were not derived from any of the experiments, rather from observed rainfall. Hence, none of the experiments was favoured over others. We have explained our procedure in Section 2.2.2, where we provide details about the spatial interpolation of the meteorological forcing.**

**Finally, the effect of calibration on the predicted flow has been highlighted adding uncalibrated flow results to Fig. 4 (such as suggested in the next comment I) and discussing them.**

E) Improved SST: The paper discusses the impact of "different accuracy levels of SST", and makes reference to "improved SST" in several places (e.g. title, abstract, section 3.1 from p8,l5, section 3.2 from p10,l6, Section 4, p15). However, the paper is critically lacking in Fig 4 and Fig 11 for example from lack of SST observations against which to conclude which systems "overestimate", "underestimate" or have "improved" SST. At best, it seems that the authors can illustrate how much the various SST products differ, but a more conclusive treatment might be possible through comparison with insitu observations (e.g. as provided through CMEMS). The authors should review the language of the argument here and throughout, but ideally provide some comparison to available in-situ SST observations.

**We agree that, without an explicit comparison with in-situ observations, it is more correct to refer simply to "higher-resolution", rather than "more accurate" or "improved" SST representation, even though in some cases (e.g., compared with the IFS SST fields along coastlines) the improvement is rather obvious. We have reviewed the language used for this topic throughout the manuscript. Nevertheless, we also have followed the interesting suggestion about a comparison of the models' skin SST fields with observations. We've made a preliminary search in the CMEMS database, in particular in the CORA database (Cabanes et al., 2013) in the latest version released (5.2, April 2019), looking for suitable observations (i.e., SST measurements at the sea-surface interface) for the dates of interest. We found some useful data, even though rather at the border of the external domain (Fig. R1; other data were available more towards the centre of the external domain and even in the innermost domain, but at depths from about 0.3 m to 0.7 m) and have provided the results of this analysis (Fig. S1 in the Supplement and related comments in the main text, P6 LL5-10, P8 LL8-12, P10 LL27-29).**

F) Case Study 1: p9, l24 – I have some issue with the conclusions to Case Study 1, and the lack of treatment of hydrological flows. Given the original nature of going from SST, to precip, to river flows within the paper, this appears to be an omission, even if the result is that there is no sensitivity in this case. Without the full 'end-to-end' treatment here, it seems confusing to justify the exploration of both Case Study 1 and Case Study 2, such that for brevity I wonder what is lost from the paper by only concentrating on Case Study 2 period. This seems to be supported by more limited assessment (e.g. no categorical statistic assessment) for Case Study 1 period. The authors should clarify this final paragraph, and better justify the need for treatment of both period 1 and 2.

**We agree with the referee that, for the sake of completeness, also Case Study 1 needs an "end-to-end" treatment, although there is no sensitivity concerning the hydrological impact in the Citrea Creek catchment. Our study shows that predictability at the scale of small/very small catchments is very limited for this kind of highly convective events, characterized by scattered and chaotic precipitation spatial patterns. We have slightly modified the related paragraph and provided details about accumulated precipitation and related peak flow in the Citrea Creek through a specific table (new Table 4).**

G) Case Study 2: I was not clear on the need for separate discussions of both 4-day and 3-day duration simulations, and again this added to the length and more confused presentation. To what extent are the authors conclusions altered by focussing only on discussion of 3-day (or 4-day) results for example? The authors should better justify the treatment of both lead times, and preferably condense the material into a more concise manuscript, perhaps highlighting the key messages of impact of lead time in the discussion. There are opportunities to consolidate some of this information within the Figures, e.g. by presenting the Taylor diagram summaries of 3-day and 4-day results alongside each other (using filled/unfilled symbols to represent different lead times perhaps).

**We first focused on the 4-day forecast to embrace the event for its entire duration. Then, we wondered about the effects of the relatively long lead time and focused also on 3-day forecasts (which, indeed, provided some differences in the hydrological impact). Following the reviewer's suggestion, Sections 3.2.1 and 3.2.2 have been merged and shortened, highlighting the key messages of the lead time impact in the discussion (P18 LL10-12). Furthermore, such as stated before (response to comment A), previous Figs. 13 and 19 have been merged, such as Figs. 17 and 20, while Fig. 18 have been moved to Supplement.**

H) It would be useful to highlight some discussion of the relative sensitivity to precipitation to SST shown here (e.g. Fig 6, Fig 12) relative to say the typical ensemble spread expected through other approaches to represent simulation uncertainty.

**We thank the referee for this useful suggestion. We have briefly discussed it, referring both to the literature and to some not yet published results (P17 LL14-19).**

I) Figure 4 – given above discussion about calibration approach, please include uncalibrated flow results here. Please also clarify the difference between Fig 4 and Fig 17 b) and d). This is another example where a more concise and consolidated presentation of results would improve the paper. For example, which experiment does the "calibrated flow" in Fig 4 show, and why are results from all experiments not included here to provide a more immediately conclusive discussion?

**As stated before (response to comment D), uncalibrated flow results have been included in Fig. 4 and discussed in the text. Furthermore, as explained in the same response, we reiterate here that calibrated data were not derived from any of the experiments, rather from observed meteorological variables. Therefore, the main difference between Fig. 4 and Figs. 17a and 17b is that the former, showing the comparison between observed hydrometric levels and simulated flow calibrated with observed precipitation, highlights how the reference discharge is retrieved for further analyses and its behaviour against available observations; the latter, once such reference discharge is established, highlights how much the forecasted discharges with the modelling chains differ from it. Though we acknowledge that the reference discharge is shown in both figures, we believe that this "double" comparison makes the paper more intelligible, leaving the reader first focus on calibration issues, then on the uncertainty of the hydrological output due to the GCM and SST boundary conditions. Therefore, for the sake of clarity, we would like to leave it.**

J) Figure 6 – it is hard to distinguish the gauge site information in the shaded circles. Consider making these larger, perhaps as unshaded circle outlines, or providing in a separate sub-plot of gauge-only observations. Again, some overlap of information with Fig 1g here.

**We agree that gauge site information here is redundant. The shaded circles have been removed. On the other hand, gauge sites in Figs. 1 have been better highlighted, enlarging both their symbols and the map (as we stated in our response to comment B). Fig. 6 has been improved adding a map of merged precipitation on the land (Sinclair and Pergram, 2005), for facilitating the quantitative comparison between forecasted and observed rain fields on the land (please, refer again to our response to comment B).**

K) Figure 8 – consider removing, given information contained in Fig 9.
**We agree with the referee. Figure 8 has been provided in the Supplemental.**

L) Figure 12 – it is confusing to have different treatments of the observed precip for different cases and across different figures, e.g. radar and gauge plots in Fig 1 and Fig 2 (quite hard to see gauge

sites in small panels), gauge points only in Fig 6, merged gauge/radar in Fig 12a. Authors should aim for more consistent treatment, perhaps across fewer figures, to provide a clearer discussion.

**We agree that maps of observed precipitation need more consistent treatment across the paper. As outlined in the previous answers (particularly to comments B and J), our strategy is the following:**

**- Revised Figs. 1 and 2 show rain gauges data overlying radar information. We opt for this solution because we believe, in the context of a general description of the events, that this is the best way to provide both quantitative information of precipitation at specific points on the land and qualitative information about the spatial precipitation patterns all over the domain (in the latter case we refer to "qualitative" information because of the radar underestimate, highlighted by the overlying of rain gauges);**

**- Revised Figs. 6 and 12 (now Fig. 11), and previous Fig. 18 now moved to Supplement, show only distributed rain fields, either simulated by the WRF configurations or reconstructed using the merging procedure proposed by Sinclair and Pergram (2005). This choice is made to favour the quantitative comparison between the amount of rain observed and simulated on the land, exploiting both the quantitative information provided by the monitoring network on the ground and the rainfall pattern description of the radar network.**

M) Figure 16 – could this be omitted (or provided in Supplementary) without any loss of understanding?
**Figure 16 has been provided in the Supplemental.**

N) Table 2 – this seems to have repeated information between the top section and 'case study 2'. I would just leave the top part for brevity and clarity.
**We thank the referee for this comment. It was an oversight, we have removed the bottom part of the Table.**

Technical corrections
=============
a) While overall the paper is well written, there are occasional places where some improvement to the English style would improve readability. As a specific example, the first sentence of Section 2.1 should be reviewed. Others might be picked up through further edit by the authors and again in the paper production process.
**The writing style has been completely re-checked.**

b) Figure 2 – caption has incorrect dates and time for this case I think.
**The referee is right. We have corrected the dates and times in the caption.**

c) P2, l5-6: Please provide specific references to the initiatives listed.
**Done (P2, LL9-10).**

d) P5, l22: Useful to translate 0.25deg. GFs resolution into an approximate grid scale length for region of interest here (i.e. how does this compare to the 16 km IFS?).
**Done (P5, L29). At these latitudes, 0.25deg is about 27 km.**

e) P8, l5: I did not follow this point – consider rewording
**Done, in the framework of an overall re-check of the writing style (P8, LL21-23).**

f) Figure 12 – typo in 'observed' on panel a)
**Corrected.**

P14, l6: Consider rewording
**The sentence has been removed.**

**References:**

[revised manuscript text omitted]